# Orai1 Boosts SK3 Channel Activation

**DOI:** 10.3390/cancers13246357

**Published:** 2021-12-17

**Authors:** Adéla Tiffner, Valentina Hopl, Romana Schober, Matthias Sallinger, Herwig Grabmayr, Carmen Höglinger, Marc Fahrner, Victoria Lunz, Lena Maltan, Irene Frischauf, Denis Krivic, Rajesh Bhardwaj, Rainer Schindl, Matthias A. Hediger, Isabella Derler

**Affiliations:** 1JKU Life Science Center, Institute of Biophysics, Johannes Kepler University Linz, A-4020 Linz, Austria; adela.tiffner@jku.at (A.T.); valentina.hopl@jku.at (V.H.); romana.schober@jku.at (R.S.); matthias.sallinger@jku.at (M.S.); herwig.grabmayr@jku.at (H.G.); carmen.hoeglinger@jku.at (C.H.); marc.fahrner@jku.at (M.F.); vici.lunz@gmx.at (V.L.); lena.maltan@jku.at (L.M.); irene.frischauf@jku.at (I.F.); 2Gottfried Schatz Research Centre, Medical University of Graz, A-8010 Graz, Austria; denis.krivic@medunigraz.at (D.K.); rainer.schindl@medunigraz.at (R.S.); 3Department of Nephrology and Hypertension, University of Bern, Inselspital, Freiburgstrasse 15, CH-3010 Bern, Switzerland; rajesh.bhardwaj@dbmr.unibe.ch (R.B.); matthias.hediger@ibmm.unibe.ch (M.A.H.); 4Department of Biomedical Research, University of Bern, Inselspital, Freiburgstrasse 15, CH-3010 Bern, Switzerland

**Keywords:** STIM1, Orai1, CRAC channel, Ca^2+^ activated K^+^ ion channels, SK channels, SK3, calmodulin, LNCaP cells

## Abstract

**Simple Summary:**

Breast, colon, and prostate cancer account for about a third of cancer cases and a fifth of cancer deaths. At the molecular level, one reason for the development of cancer is the dysfunction or altered co-regulation of cellular proteins. In this study, we focused on the co-regulation of ion channels, specifically the prominent Ca^2+^ ion channel Orai1 and the Ca^2+^ activated K^+^ ion channel SK3. It has recently been reported that their interplay promotes the growth of breast and colon cancer cells, but the molecular determinants for their co-regulation have remained elusive. In this study, we set out to characterize their interplay and the crucial regions therefore required. Moreover, we found that the function of prostate cancer cells is also controlled by the interplay of Ca^2+^ and the Ca^2+^ sensitive K^+^ channels. Our findings provide a better understanding of the co-regulation of these ion channels, which could be used in the future for the development of novel therapeutics.

**Abstract:**

The interplay of SK3, a Ca^2+^ sensitive K^+^ ion channel, with Orai1, a Ca^2+^ ion channel, has been reported to increase cytosolic Ca^2+^ levels, thereby triggering proliferation of breast and colon cancer cells, although a molecular mechanism has remained elusive to date. We show in the current study, via heterologous protein expression, that Orai1 can enhance SK3 K^+^ currents, in addition to constitutively bound calmodulin (CaM). At low cytosolic Ca^2+^ levels that decrease SK3 K^+^ permeation, co-expressed Orai1 potentiates SK3 currents. This positive feedback mechanism of SK3 and Orai1 is enabled by their close co-localization. Remarkably, we discovered that loss of SK3 channel activity due to overexpressed CaM mutants could be restored by Orai1, likely via its interplay with the SK3–CaM binding site. Mapping for interaction sites within Orai1, we identified that the cytosolic strands and pore residues are critical for a functional communication with SK3. Moreover, STIM1 has a bimodal role in SK3–Orai1 regulation. Under physiological ionic conditions, STIM1 is able to impede SK3–Orai1 interplay by significantly decreasing their co-localization. Forced STIM1–Orai1 activity and associated Ca^2+^ influx promote SK3 K^+^ currents. The dynamic regulation of Orai1 to boost endogenous SK3 channels was also determined in the human prostate cancer cell line LNCaP.

## 1. Introduction

Calcium (Ca^2+^) ionsplay a variety of critical roles in many vital aspects of cellular life. Ca^2+^ signaling triggers short- and long-term cellular processes responsible for fundamental physiological functions, including secretion, gene regulation, muscle contraction, activation of the immune system, cell proliferation, cell motility, and apoptosis [1,2,3,4,5,6,7]. Defects in the cellular Ca^2+^ homeostasis due to dysfunction or changes in Ca^2+^ signal transduction can lead to severe immune deficiencies, neurological diseases, cardiovascular problems, and various types of cancers [8]. Ca^2+^ enters the cell via diverse Ca^2+^ ion channels. The resulting elevations of intracellular Ca^2+^ levels can regulate the activity of Ca^2+^ sensing ion channels. Several reports have already provided evidence of an interplay of Ca^2+^ and Ca^2+^ regulated potassium (K^+^) ion channels such as Ca_V_2.3 with BK channels [9], Orai1 with the ether a-gogo K^+^ channel hEag1 [10] or BK channels [11,12], Orai1 [13], or TRPV6 [14] with the small conductance Ca^2+^-activated activated K^+^ channels SK4 and Orai1 with its homolog SK3 [13,15,16,17,18,19]. The interplay of the Ca^2+^ ion channel Orai1 with the Ca^2+^-activated K^+^ ion channel SK3 is currently best studied in breast cancer cells [16,20,21].

Orai1 [22,23] together with the ER-localized Ca^2+^ sensor STIM1 [24,25,26] constitute the so-called Ca^2+^ release-activated Ca^2+^ (CRAC) channel. CRAC channels are activated upon ER Ca^2+^ store depletion [22,27]. In response to the drop of ER Ca^2+^ concentration, STIM1 undergoes a conformational change [25,28,29,30,31,32], coupling to and activating Orai1 to allow Ca^2+^ influx [33,34,35,36]. STIM1 induced Orai1 activation is only guaranteed as long as the cytosolic N- and C-terminus, as well as the loop2 region of this Ca^2+^ ion channel, are intact [30,37,38,39,40,41,42,43,44,45,46]. STIM1–Orai1 coupling, predominantly established via the main coupling site at the Orai1 C-terminus, can be disrupted via either a single point mutation L273D or C-terminal truncations to L276 or beyond [35,38,47,48]. N-terminal truncations and site-directed mutagenesis revealed that residues downstream of Q72 are indispensable for full activation of STIM1 mediated Ca^2+^ entry [37,38,47]. The residues 39–59 of the Orai1 N-terminus associate with the A-kinase anchoring protein (AKAP79) that binds both calcineurin and nuclear factor of activated T-cells (NFAT). Orai1-AKAP79 association thus allows local calcineurin activation and rapid NFAT nuclear translocation in response to local Ca^2+^ entry through Orai1 [49].

SK3 channels are part of the SK channel family consisting of the small (SK1, SK2, and SK3) and intermediate (SK4) conductance Ca^2+^-activated K^+^ channels. The pore forming α subunits of SK channels consist of a tetrameric six-transmembrane domain (S1–S6) structure flanked by the cytosolic N- and C-terminal strands. The S4–S5 linker in the SK channel contains two α-helices, S_45_A, and S_45_B. The SK channel gating is triggered by changes in intracellular Ca^2+^ concentrations at the submicromolar level (K_D_ = 0.5 µM) [50] via bound calmodulin (CaM) and occurs in a voltage-independent manner [51,52]. On the basis of the structural and functional analyses, Lee et al. [53] presented the Ca^2+^-dependent channel opening mechanism of SK4. The high structural similarity between SK3 and SK4 suggests that, in analogy to SK4, similar Ca^2+^-dependent pore opening mechanisms exist for SK3. As shown by Lee et al. [53], four CaM molecules bind to each channel tetramer. The C-terminal lobe of CaM is constitutively bound to SK4 while the N-terminal lobe binds SK4 in a Ca^2+^ dependent manner, thereby controlling channel opening. Elevations of intracellular Ca^2+^ levels allow Ca^2+^ binding to the CaM N-lobe, triggering a conformational change of the S4–S5 linker, which results in pore opening [50,53]. Ca^2+^ independent CaM binding has been further reported to regulate SK2 channel trafficking to the membrane [54].

The structure–function relationship of Orai1 and SK3 is currently well understood.

Recent studies have reported the co-localization of the SK3 and Orai1 channels in breast and colon cancer cells [13,16,18,19,55,56,57]. There, these channels control cell proliferation and migration or trigger bone metastasis [15,16,19,56,58,59], as found, for example, for SK3–Orai1 complexes in breast cancer cells [56]. While normal human breast and colon cancer cells contain only Orai1, the corresponding cancer cells express SK3 and Orai1 co-localized in lipid rafts [16]. Those cells exhibit constitutive Orai1-dependent Ca^2+^ entry, independent of STIM1, which is promoted via hyperpolarization due to SK3 channel activation. This has led to the assumption that Orai1-induced SK3 activation is mediated via direct or indirect interaction of those two proteins, potentially inducing structural changes to open the Orai pore subsequently, leading to the constitutive activity of the channel [57]. Indeed, there is evidence for a co-regulation of SK3 and Orai1 in breast [17] and colon [14,16,19,60] cancer cells. However, the structural requirements and key determinants manifesting their interplay are still unknown.

Here, we present crucial, previously elusive conditions that mediate the communication between the SK3 and Orai1 channels. We show that Orai1 promotes SK3 K^+^ currents and restores SK3 channel activity, which has been abolished by CaM mutants. Using a set of point and truncation mutants in Orai1, we uncovered that an intact pore geometry and virtually the entire Orai1 N- and C-termini are crucial for the SK3–Orai1 interplay. While the SK3–Orai1 co-regulation occurs independently of STIM1, STIM1 is able to reduce the extent of SK3–Orai1 co-localization, and thus their interplay under physiological conditions. The SK3–Orai1 co-regulation occurs not only in the standard HEK 293 expression system, but also in the human prostate cancer cell line, LNCaP.

## 2. Materials and Methods

### 2.1. Reagents

Inhibitors (NS8593 hydrochloride Cat #: N-195, 4-AP Cat #: A-115) and activators (NS-309 Cat #: N-180, 1-EBIO Cat #: E-150, Cyppa Cat #: C-110) were purchased from Alomone Labs (Jerusalem, Israel).

### 2.2. Molecular Biology

For N-terminal fluorescence labeling of human Orai1 (Orai1; Accession number NM_032790, kindly provided by A. Rao’s lab (Center for Autoimmunity and Inflammation, La Jolla Institute for Immunology, UC San Diego’s Research Park, San Diego, CA, USA), the constructs were cloned into the pEYFP-C1 (Clontech Laboratories, Inc. (since 2015 Takara Bio USA, Inc. San Jose, CA, USA)) expression vector via KpnI and XbaI (Orai1) restriction sites, respectively. Orai1 N-terminal deletion mutants amplified via PCR including a 5′ KpnI and a 3′ XbaI restriction site for cloning into the pEYFP-C1 vector as described in [33]. Site-directed mutagenesis of Orai1 was performed using the QuikChange™ XL site-directed mutagenesis kit (Stratagene (since 2007 Agilent Technologies, Inc., San Diego, CA, USA) with the corresponding Orai1 constructs serving as a template according to [53].

For N-terminal fluorescence labeling of human SK3 (SK3; Accession number AAP45947.1, kindly provided by N. Marrion’s lab (School of Physiology, Pharmacology & Neuroscience, University of Bristol, Bristol, England), the constructs were cloned into the pEYFP-C1 (Clontech Laboratories, Inc.) expression vector via EcoRI and BamHI (SK3) restriction sites, respectively.

Human STIM1 (STIM1; Accession number: NM_003156) N-terminally ECFP-tagged was kindly provided by T. Meyer’s Lab (Chemical and Systems Biology, Stanford University, Stanford, CA, USA).

CaM was amplified by PCR and cloned into YFP-/CFP-C1 vectors using EcoRI and SacII. Frame correction was performed at the beginning of the MCS, leading to a frame equivalent to YFP-/CFP-C2 vectors.

The CaM mutant (CaMMUT) has mutations introduced on all four EF hands, rendering it insensitive to Ca^2+^. Point mutations within CaM have been performed using the QuikChange™ XL site-directed mutagenesis kit (Stratagene).

All clones were confirmed by sequence analysis.

#### 2.2.1. Generation of STIM1/Orai1 Double Knockout HEK 293 Cells Using CRISPR/Cas9 Genome Editing

The guide RNA pairs within exon 1 of STIM1 were identified using the Benchling CRISPR webtool (https://benchling.com/crispr, accessed on 12 October 2015) and the following complementary oligos with BbsI compatible overhangs were designed:STIM1_gRNA1 for: 5′-CAC CGT TCT GTG CCC GCG GAG ACT C-3′STIM1_gRNA1 rev: 5′-AAA CGA GTC TCC GCG GGC ACA GAA C-3′STIM1_gRNA2 for: 5′-CAC CGT ATG CGT CCG TCT TGC CCT G-3′STIM1_gRNA2 rev: 5′-AAA CCA GGG CAA GAC GGA CGC ATA C-3′

The oligo pairs were annealed by incubating for 5 min at 95 °C in a thermocycler and then ramping down to 25 °C at 5 °C/min. The dsDNA guide inserts were ligated into BbsI-digested pX330-PGK-puro vector. HEK 293 cells seeded in a 6-well plate were co-transfected with 1 μg of each plasmid using Lipofectamine^®^ 3000 (Thermo Fisher Scientific, Waltham, MA, USA) when the cells were 50–60% confluent. The transfectants were subjected to 96 h of puromycin selection (1.5 μg/mL), 24 h after transfection. For further knockout of Orai1, the guide RNA pairs within exon 2 and exon 3 were identified using the Benchling CRISPR webtool and the following complementary oligos with BbsI compatible overhangs were designed:Exon 2:
Orai1_gRNA1 for: 5′-CACCGATCGGCCAGAGTTACTCCG-3′Orai1_gRNA1 rev: 5′-AAACCGGAGTAACTCTGGCCGATC-3′Exon3:
Orai1_gRNA2 for: 5′-CACCGGCGGAGTTTGCCCGCTTAC-3′Orai1_gRNA2 rev: 5′-AAACGTAAGCGGGCAAACTCCGCC-3′

The oligo pairs were annealed as mentioned above. The dsDNA guide inserts with BbsI overhangs were ligated into BbsI-digested pU6-(BbsI) CBh-Cas9-T2 A-mCherry (Addgene (Watertown, MA, USA) Plasmid #64324). The resulting constructs were co-transfected in HEK 293 STIM1 KO cells seeded in a T25 flask using Lipofectamine 2000. The transfected cells were FACS-sorted with mCherry fluorescence, and single cells were seeded in a 96-well plate. The cells were tested by Ca^2+^ imaging with FURA-2 and a single-cell-derived clone was functionally confirmed to have undergone knockout of STIM1 and Orai1. Cells were used until passage 18.

#### 2.2.2. Western Blot Analysis and Co-Immunoprecipitation

Untransfected as well as transfected (transfection 16–24 h prior to lysis) wild-type or CRISPR/Cas9 STIM1/Orai1 double knockout (STIM1/Orai1 DKO) HEK 293 cells were cultured in 12 cm dishes, harvested, and washed twice in an HBSS (Hank’s balanced salt solution) buffer containing 1 mM EDTA. After centrifugation (1000× *g*/2 min), cell pellets were resuspended in homogenization buffer (25 mM Tris HCl pH 7.4, 50 mM NaCl, protease inhibitor (Roche, Basel, Switzerland)) and incubated on ice for 15 min. Lysed cells were passed 10 times through a 27G ½″ needle and centrifuged at 1000× *g* for 10 min at 4 °C to pellet debris. Then, 21 µL of each sample was mixed with nonreducing Laemmli’s buffer, heated for 10 min at 55 °C, and subjected to 3–8% Tris-acetate gels (BioRad, Vienna, Austria). Samples were loaded on a 12% SDS page, transferred to a nitrocellulose membrane, and immunoblotted with suitable antibodies (antiSK3 antibodies (Alomone labs), antiSTIM1 (CellSignaling, Frankfurt am Main, Germany), antiOrai1, antirabbit, and antiActin (SigmaAldrich, Taufkirchen, Germany)).

For co-immunoprecipitation, detached cells were incubated in lysis buffer for 2 h on ice with constant shaking. After centrifugation (1000× *g*, 10 min) and collecting the supernatant, 3 µg of anti-Orai1 or anti-STIM1 antibodies were added and shaken at 4 °C over night. On the next day, 100 µL Sepharose A beads were added and incubated for 2 h at 4 °C with constant shaking. Samples were centrifuged for 10 s at maximum speed and beads washed 3× with ice cold lysis buffer. 50 µL of 2× Laemmli buffer were added to the beads and boiled for 5 min at 95 °C. Samples were then loaded on a 12% SDS PAGE, blotted, and detected with an anti-SK3 antibody (Alomone labs). A Precision Plus Dual Color standard was used for size comparison (BioRad).

Each experiment was performed at least 3 independent times.

### 2.3. Cell Culture and Transfection

The transient transfection of HEK 293 or STIM1/Orai1 DKO HEK 293 cells was performed [61] using the TransFectin Lipid Reagent (BioRad) with the corresponding plasmids. The transient transfection of passages 3–5 of LNCaP cells was performed using the Lipofectamine^®^ 3000 transfection Reagent (Thermo Fisher Scientific) with the corresponding plasmids. Measurements were carried out 24 h following transfection. Notably, all LNCaP cell experiments described here were feasible only within the cell passages 3–5. Earlier and later passages failed to reveal SK3 channel activation (see Section 3.6) upon the same solution conditions.

### 2.4. Electrophysiology

Electrophysiological recordings that assessed the characteristics of 2–3 constructs were carried out in paired comparison on the same day. Expression patterns and levels of the various constructs were carefully monitored by confocal fluorescence microscopy and were not significantly changed by the introduced mutations. Electrophysiological experiments were performed at 20–24 °C, using the patch–clamp technique in the whole-cell recording configuration. For SK3/Orai1 current measurements, voltage ramps were usually applied every 5 s from a holding potential of 0 mV, covering a range of −90 to +90 mV over 1 s. Used solution conditions to characterize STIM1/Orai1 and SK3 currents individually, but also in their interplay, are systematically summarized in Appendix B. The two main solution conditions applied are Physiological SK3 Solution Conditions with [EGTA]_intra_ and Symmetrical SK3 Solution Conditions with [EGTA]_intra_. They share the same intracellular pipette solution but differ in the composition of the extracellular solution. The internal pipette solution contained (in mM) 144 KCl, 1.49 MgCl_2_, 10 HEPES, and 0.1 EGTA (or 0.1 BAPTA). The physiological extracellular solution consisted of (in mM) 140 NaCl, 5 KCl, 1 MgCl_2_, 10 HEPES, 10 glucose, and 2 CaCl_2_, with a pH of 7.4. The corresponding symmetrical extracellular solution consisted of (in mM) 144 KCl, 1 MgCl_2_, 10 HEPES, 10 glucose, and 10 CaCl_2_, with a pH of 7.4. The currents recorded under physiological solution conditions were obtained at +30 mV, while the ones recorded under symmetrical solution conditions were obtained at −74 mV. Furthermore, Orai1 Ca^2+^ currents were obtained at −74 mV. The Na^+^-DVF solution contained 150 mM NaCl, 10 mM HEPES, 10 mM glucose, and 10 mM EDTA, with a pH of 7.4. Applied voltages were not corrected for the liquid junction potential, which was determined as +12 mV. Experiments were performed using HEK 293 cells, STIM1/Orai1 DKO HEK 293 cells, or LNCaP cells as indicated.

### 2.5. Confocal Fluorescence Microscopy

Confocal FRET microscopy was performed on normal HEK 293 or CRISPR/Cas9 STIM1/Orai1 double knockout (STIM1/Orai1-DKO) HEK 293 cells. The transfected cells were grown on coverslips for 24 h and subsequently transferred to an extracellular solution consisting of 140 mM NaCl, 5 mM KCl, 1 mM MgCl_2_, 2 mM CaCl_2_, 10 mM glucose, and 10 mM HEPES buffer (adjusted to pH 7.4 with NaOH). The experimental setup consisted of a CSU-X1 Real-Time Confocal System (Yokogawa Electric Corporation, Musashino, Tokyo, Japan) combined with two CoolSNAP HQ2 CCD cameras (Photometrics, Tucson, AZ, USA). The installation was also fitted with a dual port adapter (dichroic, 505lp; cyan emission filter, 470/24; yellow emission filter, 535/30; Chroma Technology Corporation, Olching, Germany). An Axio Observer.Z1 inverted microscope (Carl Zeiss, Oberkochen, Germany) and two diode lasers (445 and 515 nm, Visitron Systems, Puchheim, Germany) were connected to the described configuration. All described components were positioned on a Vision IsoStation antivibration table (Newport Corporation, Irvine, CA, USA). Image recording and control of the confocal system were carried out with the VisiView software package (v.2.1.4, Visitron Systems). Cross-excitation and spectral bleed-through necessitate image correction before any FRET calculation. Cross-excitation calibration factors were therefore determined for all expressed DNA constructs on each measurement day. After threshold determination as well as background signal subtraction, the apparent FRET efficiency Eapp was calculated on a pixel-to-pixel basis. This was performed with a custom program48 integrated into MATLAB (v.7.11.0, The MathWorks, Inc., Natick, MA, USA) that implements a microscope-specific constant G parameter of 2.75.

Fluorescence images of the subcellular localization of NFAT transcription factors in HEK 293 cells were recorded using a QLC100 Real-Time Confocal System (VisiTech Int., Sunderland, UK) connected to two Photometrics CoolSNAPHQ monochrome cameras (Roper Scientific, Planegg, Germany) and a dual-port adapter (dichroic: 505lp; cyan emission filter: 485/30; yellow emission filter: 535/50; Chroma Technology Corp., Olching, Germany). This system was attached to an Axiovert 200M microscope (Zeiss, Oberkochen, Germany) in conjunction with two diode lasers (445 nm, 515 nm) (Visitron Systems, Puchheim, Germany). Visiview 2.1.1 software (Visitron Systems) was used for image acquisition and control of the confocal system. ImageJ was employed for subcellular localization analysis of the transcription factors by means of intensity measurements of the cytosol and nucleus, distinguishing between three different populations with different nucleus/cytosol ratios: inactive (<0.85), homogenous (0.85–1.15), and active (>1.15).

### 2.6. Fluorescence-Based Ca^2+^ Imaging

STIM1/Orai1 DKO HEK 293 cells were loaded with 1 mM Fura-2 AM for 20 min at 37 °C in Ringer solution containing 145 mM NaCl, 5 mM KCl, 10 mM glucose, 10 mM HEPES, and 1 mM MgCl^+^, 2 mM CaCl for 2 mM Ca^2+^. The cells were then washed three times, and the coverslips were mounted on an Axiovert 135 inverted microscope (ZEISS, Oberkochen, Germany), where fluorescence was recorded from individual cells, with excitation wavelengths of 340 and 380 nm and emission wavelength at 505 nm. Changes in Ca^2+^ were monitored using the Fura-2 340/380 fluorescence ratio and calibrated according to the method established by Grynkiewicz et al. [48]. The fluorescence microscope was equipped with a monochromator (T.I.L.L. Photonics, Kaufbeuren, Germany) and corresponding filter sets and allowed the detection of CFP/YFP/red fluorescent protein fluorescence.

### 2.7. Co-Localization Analysis

The same technical equipment as for confocal FRET microscopy was used. Co-localization analysis was carried out when images of a co-expression indicated that two proteins were localized at the same positions (pixel-by-pixel analysis, orange in the CFP/YFP overlay image). The Pearson correlation coefficient (*R*-value) was used to quantify the strength of the co-localization, where a value of *R* = 1 signifies perfect positive correlation/co-localization.

### 2.8. Cell Proliferation Assay

Cells were seeded at an initial density of 1.5 × 10^4^ cells/well in 24-well plates (ThermoFisher). Cells were incubated for 4 days in a medium (control) or with a medium, containing agonists (NS309, Cyppa) or antagonists (NS8593 hydrochloride, 4-AP, La^3+^, Synta66, GSK7975A, BTP2) of the SK3 channel or Orai1 channel at various concentrations, respectively. The CellTiter 96^®^ AQueous Non-Radioactive Cell Proliferation Assay kit commercially available (Promega, Walldorf, Germany), containing MTS solution (tetrazolium compound) and PMS solution (electron coupling reagent), was used to calculate the cell growth upon different conditions. Cells treated with MTS/PMS solution were equilibrated for 4 h at 37 °C incubator before the measurement of absorption at 490 nm by plate reader Zenyth3100 was executed. Every measurement was performed in triplicate.

### 2.9. Statistics

All data are presented as the mean ± SEM (standard error of the mean) for the indicated number of experiments. Statistical significance was determined by Mann–Whitney test for comparison of two groups (using Origin Pro 2019, Northampton, MA, USA). Statistical significance was set to *p* < 0.05 and is indicated in the bar diagrams with an asterisk (*).

## 3. Results

### 3.1. Mild Cytosolic Ca^2+^ Buffering Allows Robust SK3 Channel Activation and Weak STIM1-Orai1 Activation

Previous reports on breast and colon cancer cells revealed enhanced Ca^2+^ levels due to an interplay of SK3 and Orai1, independent of STIM1 [16,17]. Prior to our investigations of a potential co-regulation of SK3 and Orai1 channels in HEK 293 cells, we systematically examined individual SK3 and STIM1-Orai1 current activation upon their heterologous expression, using distinct solution conditions, differing mainly in intracellular Ca^2+^ and/or EGTA concentration. The detailed solution compositions used are highlighted by italics/underscore and summarized in Appendix B. Our goal was to confirm the Ca^2+^ sensitivity of SK3 channels and to reach intracellular Ca^2+^ concentrations, which still allow the activation of SK3 currents, but leave STIM1 and Orai1 in the resting or marginally active state.

As shown in previous reports [62,63,64], we confirmed via electrophysiological experiments that SK3 K^+^ currents enhance with increasing intracellular Ca^2+^ concentrations. For this purpose, we used both symmetrical and physiological solution conditions with the pipette solution containing progressively increasing buffered Ca^2+^ concentrations (50, 100, 250, 350, 600, 800, and 1000 nM; symmetrical or physiological SK3 solution conditions with buffered [Ca^2+^]_intra_). SK3 evoked K^+^ currents, as determined using repetitive voltage-ramps from −90 to +90 mV, exhibited under symmetrical SK3 K^+^ solution conditions a double rectifying I/V relationship and a reversal potential of ~0 mV [62,63,64] (Appendix A). Under physiological SK3 K^+^ solution conditions, SK3 currents displayed an outward rectifying current with a reversal potential of ~ −70 mV, in accord with Xia et al. [64] (Appendix A). Under both conditions, increasing Ca^2+^ concentrations in the pipette solution resulted in enhanced SK3 mediated K^+^ currents (Appendix A).

Additionally, we revealed via varying EGTA concentrations in the pipette solution (symmetrical or physiological SK3 solution conditions with [EGTA]_intra_) that 100 µM EGTA enabled the activation of SK3 mediated K^+^ currents (Figure 1A–C and Appendix A). SK3 K^+^ currents exhibited upon whole-cell break-in slight constitutive activity, which reached a steady-state level after ~250 s, as determined from repetitive voltage-ramps at voltages with highest currents reached (−74 mV for symmetrical and +30 mV for physiological solution conditions; Figure 1A−C and Appendix A). Higher EGTA concentrations in the pipette solution, which decrease intracellular Ca^2+^ concentrations, strongly reduced (200 µM EGTA) or abolished (>200 µM EGTA) SK3 K^+^ current activation (Figure 1A–C and Appendix A). Thus, cytosolic Ca^2+^ levels reached with 100 µM EGTA appear to be sufficient for SK3 channel activation, while higher EGTA buffers decreased cytosolic Ca^2+^ too much to allow SK3 activation. The I/V relationships obtained using 100 µM EGTA for both, physiological as well as symmetrical solutions, are comparable to those obtained with increasing intracellular Ca^2+^ concentrations (symmetrical or physiological SK3 solution conditions with buffered [Ca^2+^]_intra_; Figure 1C and Appendix A).

Furthermore, we determined whether the above-mentioned solution conditions: physiological SK3 solution conditions with [EGTA]_intra_, used to characterize SK3 channel activation, allow STIM1-Orai1 activation. Using physiological SK3 solution conditions with 100 µM EGTA in the pipette solution led to weak STIM1-Orai1 activation, which enhanced upon the exchange of the extracellular [Ca^2+^] from 2 to 10 mM (physiological SK3 solution conditions with high [Ca^2+^]_extra_) (Figure 1D,E). Consistent with these findings, STIM1-Orai1-Ca^2+^ current activation was five-fold weaker in the presence of 100 µM EGTA (standard STIM1/Orai1 solution conditions with low [EGTA]_intra_) than with 20 mM EGTA in the pipette solution (standard STIM1/Orai1 solution conditions), both in the presence of 10 mM Ca^2+^ extracellularly (Figure 1F,G).

Overall, we confirmed that SK3 channel currents activate increasingly with elevated intracellular Ca^2+^ concentrations, both via increasing buffered Ca^2+^ concentrations as well as decreasing EGTA concentrations in the pipette solution. It turned out that 100 µM EGTA in the pipette solution and 2 mM Ca^2+^ in the extracellular solution are suitable to further investigate the SK3–Orai1 interplay, as they allowed robust SK3 channel activation, but weak STIM1/Orai1 activation. Thus, we used physiological SK3 solution conditions with [EGTA]_intra_, in particular with 100 µM EGTA intracellularly and 2 mM Ca^2+^ extracellularly in the following experiments to determine SK3 K^+^ currents and its potential interplay with Orai1.

### 3.2. Orai1 Boosts SK3 Channel Activation

In this part of the study, we investigated whether a potential interplay of SK3 and Orai1, as it occurs in specific cancer cell types [16,17], also applies to HEK 293 cells in the absence of STIM1.

Initially, we examined whether co-expression of Orai1 and SK3 in HEK 293 cells affects intracellular Ca^2+^ levels, as suggested for breast cancer cells [16,17]. Under physiological buffer conditions, we observed marginal, but not significant enhancement of basal Ca^2+^ levels with Fura-AM in SK3 and Orai1 co-expressing cells compared to only Orai1 or SK3 containing cells (Appendix A). Ca^2+^ mediated NFAT activation was only slightly, but not significantly enhanced for SK3–Orai1 compared to only Orai1 or SK3 expressing cells (Appendix A). From these experiments, we assume that there is only a minor rise in global cytosolic Ca^2+^ levels under physiological buffer conditions.

In the next step, we investigated whether Orai1 and SK3 co-expressing cells exhibit constitutive Ca^2+^ currents. Interestingly, using standard STIM1/Orai1 solution conditions with low [EGTA]_intra_ buffering in electrophysiological experiments, SK3 and Orai1 co-expressing cells exhibited robust constitutive current activation (Figure 1H–K) with a double rectifying I/V relationship with a shape similar to SK3 K^+^ currents obtained under physiological conditions. These currents were strongly reduced in the absence of Orai1 (Figure 1K and Appendix A). Instead of using 100 µM EGTA, 20 mM EGTA in the pipette solution abolished these currents as well (standard STIM1/Orai1 solution conditions; Figure 1H,I). Application of NS8593, selectively interfering with SK1, SK2, and SK3 [65], significantly diminished currents developed in SK3 expressing cells in the presence of low [EGTA]_intra_ (Figure 1J,K). For solely SK3 expressing cells, K^+^ currents were fully inhibited upon NS8593 application (Figure 1K and Appendix A). Remarkably, for SK3–Orai1 expressing cells, a small, widely inward rectifying current with a reversal potential of +40 mV remained (Figure 1J and Appendix A). This remaining current was completely blocked via 10 µM La^3+^ (Figure 1J). The latter indicates that, contrary to SK3 expressing cells, Orai1-SK3 co-expressing cells show not only K^+^ permeation, but also Ca^2+^ influx, potentially via Orai1, in line with previous findings [16,17]. Moreover, our observations (Figure 1I–K) point to a potentiating effect of Orai1 on SK3 K^+^ currents.

Thus, in the following, we investigated, using defined symmetrical and physiological SK3 solution conditions with [EGTA]_intra_, whether SK3 K^+^ currents are altered by the co-expression of Orai1. Indeed, SK3 K^+^ currents were twice as high in the presence of Orai1 compared to its absence under both solution conditions (Figure 2A–C and Appendix A). It is noteworthy that in the absence of extracellular Ca^2+^ (0 mM Ca^2+^) no SK3 K^+^ currents are activated. Only upon the exchange to a 2 mM Ca^2+^ containing bath solution were SK3 K^+^ currents activated, which further increased in the presence of 10 mM Ca^2+^ (Figure 2D).

To examine whether endogenous STIM1 and Orai1 are involved in the Orai1 mediated boost of SK3 K^+^ currents, we performed the above-described experiments analogously in CRISPR/Cas9 STIM1/Orai1 double knockout (STIM1/Orai1-DKO) HEK 293 cells verified by sequence analysis and Western blot (Appendix A). In accordance with our observations in normal HEK 293 cells, we discovered significantly enhanced SK3 K^+^ currents in the presence compared to the absence of Orai1 (Figure 2E–G). In addition to the loss of SK3 K^+^ currents in the absence of extracellular Ca^2+^ (0 mM Ca^2+^), and the sequential enhancement in SK3 K^+^ currents with increasing extracellular Ca^2+^ concentrations (Figure 2D), the use of a divalent free Na^+^ containing solution, known to permeate Orai1 channels [45,66,67,68,69], did not generate SK3 K^+^ current activation (Appendix A).

In support of these findings, not only the application of the general Ca^2+^ ion channel blocker La^3+^ (10 µM), but also of the CRAC channel blocker GSK-7975A (10 µM) to cells expressing SK3 and Orai1 led to a reduction in K^+^ currents similar to those obtained in SK3 containing cells in the absence of Orai1 (Figure 2B,F and Appendix A). Moreover, SK3 K^+^ currents enhanced by the SK channel agonist Cyppa [62] were significantly inhibited by La^3+^ only in the presence, but not in the absence of Orai1 (Figure 2F). Additionally, in contrast to Orai1, the dominant-negative Orai1 pore mutant, Orai1 E106Q, when co-expressed with SK3, did not further stimulate SK3 mediated K^+^ currents (Figure 2A–C,E–G and Appendix A). This suggests that enhancements of SK3 K^+^ currents in the presence of Orai1 are likely mediated via local Ca^2+^ entry across Orai1.

The SK3 current enhancing role of Orai1 is also evident when the intracellular Ca^2+^ concentration is varied using different EGTA concentrations. In the absence of Orai1, SK3 K^+^ currents were readily observed in the presence of up to 100 μM EGTA in the pipette and they significantly increased by the presence of Orai1. At higher (200–500 μM) EGTA concentrations, however, activation of K^+^ currents was only observed in the presence of Orai1. With 1 mM EGTA in the pipette, no SK3-mediated K^+^ currents were detectable, regardless of whether Orai1 was present or absent (Figure 2H and Appendix A). Moreover, when using 100 µM BAPTA instead of 100 µM EGTA, we discovered that SK3 K^+^ currents were not further enhanced in the presence of Orai1 compared to its absence (Figure 2H and Appendix A).

It is worth mentioning that we verified overexpression of CFP-labelled SK3 and YFP-Orai1 upon the individual or upon co-expression using confocal fluorescence microscopy (Appendix A) as well as Western blot analysis in wild-type and STIM1/Orai1 DKO HEK 293 cells. (Figure 2J, Appendix A). Interestingly, single transfection of SK3 enabled only the detection of higher order oligomers (Appendix A), while triple expression of SK3, Orai1, and STIM1 enabled the predominant detection of SK3 monomers (Appendix A).

Moreover, we discovered clear co-localization of Orai1 and SK3 in specific spots of the cell membrane of HEK 293 cells, in line with previous reports showing their co-localization in breast cancer cells [16] (Figure 2I). Analysis of SK3 and Orai1 E106Q revealed a comparable level of co-localization (Figure 2I). Thus, reduced K^+^ current densities of SK3 in the presence of Orai1-E106Q are most likely the result of abolished Ca^2+^ entry via the Orai1 pore mutant per se. In line with the co-localization of SK3 and Orai1, we discovered co-immunoprecipitation of SK3 and Orai1 in wild-type HEK 293 and HEK 293 STIM1/Orai1 DKO cells (Figure 2J and Appendix A), which is occurring to a much lower extent in STIM1/Orai1 DKO HEK 293 cells co-expressing SK3 and STIM1 (Figure 2J and Appendix A).

As we obtained in both normal and STIM1/Orai1 DKO HEK 293 cells comparable results, we performed our subsequent experiments in STIM1/Orai1 DKO HEK 293 cells only.

Moreover, we engineered an SK3 pore mutant, SK3 V544W, based on the cryo-EM structure of SK4 [53] (Appendix A—sequence alignment). The substitution of valine to tryptophan at the narrowest part of the cytosolic gate [53] kept this mutant almost inactive. The expression of SK3 V544W and its co-localization with Orai1 remained comparable to that of SK3 (Appendix A). Remarkably, while SK3 V544W showed significantly reduced currents compared to SK3 wild-type, Orai1, but not Orai1 E106Q, partially restored its activity (Appendix A).

In summary, we clearly showed that Orai1- and SK3- co-expressing cells exhibit constitutive K^+^ currents. Inhibition of SK3 K^+^ currents left a tiny, mainly inward rectifying current with a V_rev_ in the range of +40 mV, which could be blocked by La^3+^. This suggests that SK3 and Orai1 co-expressing cells develop marginal, local increases in intracellular Ca^2+^ levels, which is sufficient to boost SK3 K^+^ currents. Orai1-induced SK3 K^+^ current activation in HEK 293 cells is accompanied by their close co-localization. Enhancements of SK3 K^+^ currents are specifically driven by Orai1, as this is impaired by the expression of the prominent Orai1 E106Q pore mutant as well as by the application of a CRAC channel blocker. In addition, the activity of an SK3 pore mutant could be partially restored by Orai1, suggesting that the close co-localization of SK3 and Orai1 allows allosteric interference with the SK3 pore opening mechanism.

### 3.3. Orai1 Overrules the Inhibitory Effect of CaM Mutants on SK3 Channels

While we reported here that Orai1 boosts SK3 K^+^ currents, it is believed that SK channels are generally activated via CaM. Extensive electrophysiological and pull-down studies on SK2 channels, investigation of the effect of disease-related CaM mutants on SK3 [70] and recent structural resolutions on the SK4–CaM complex revealed clear evidence that their activation is driven by Ca^2+^ bound CaM [50,51,53,54,64,71]. Due to the similar Ca^2+^ dose–response relationship of all SK channels and their conserved structural CaM binding domains, they are all assumed to be activated by CaM [53,54,71]. In line with previous studies on SK channels [50,51,53,54,70,71], we provide using physiological SK3 solution conditions with [EGTA]_intra_ experimental evidence that SK3 channels are regulated in a CaM dependent manner analogously. While the co-expression of CaM_WT_ with SK3 leads to significantly enhanced K^+^ currents (Appendix A), the presence of the CaM_1,2,3,4_ (CaM_MUT_) mutant, deficient in Ca^2+^ binding at all four EF-hands, abolished SK3 K^+^ currents almost completely (Figure 3A,B and Appendix A).

In the following, we investigated the effects of the CaM_1,2_ and CaM_3,4_ mutants deficient in Ca^2+^ binding to the N- and C-lobe, respectively. While CaM_1,2_ impaired SK3 mediated K^+^ currents to a similar extent to CaM_MUT_ (Figure 3C,D), CaM_3,4_ led to similar enhancements in SK3 K^+^ currents such as CaM_WT_ (Appendix A). Moreover, we detected significant FRET and distinct co-localization for CFP- labeled SK3 and YFP-labeled CaM proteins and mutants confirming their direct interaction (Figure 3G,H and Appendix A). This suggests, in accordance with the previous reports on SK channels [50,51,53,54,64,71], that CaM is constitutively bound to the SK3 channel and upon Ca^2+^ binding to its N-lobe it triggers conformational changes leading to pore opening.

We further investigated novel SK3 mutants assumed to be defective in CaM mediated activation, based on SK4 studies [53]. Indeed, SK3 S441E and SK3 S441W, containing the binding site for CaM N-lobe mutated, exhibited no activation upon co-expression with CaM or CaM_3,4_ (Figure 3E,F and Appendix A). Localization of these SK3 mutants and their FRET with CaM were maintained to a comparable extent as for SK3 wild-type (Figure 3I and Appendix A).

In the following, we continued to examine the effect of Orai1 on the interplay of SK3 and CaM. While co-expression of either CaM or Orai1 together with SK3 led to significantly increased K^+^ currents, triple co-expression of SK3, CaM, and Orai1 further increased the currents only slightly. (Figure 3E and Appendix A). Thus, co-expression of Orai1 is sufficient to obtain almost maximum SK3 channel activity as obtained in the presence of CaM.

Interestingly, a co-expression of SK3, Orai1, and a CaM mutant (CaM_MUT_, CaM_1,2_) led to significantly enhanced K^+^ currents compared to the abolished SK3 activity in the presence of a CaM mutant (CaM_MUT_, CaM_1,2_) without Orai1 (Figure 3A–D). The defective Orai1 E106Q was unable to overrule the inhibitory effect of the CaM_MUT_ on SK3 (Appendix A). In line with these findings, a co-expression of SK3 CaM-binding-site mutants, SK3 S441E or SK3 S441W with Orai1 left the mutated channel inactive, both in the absence and presence of CaM (Figure 3E,F and Appendix A).

To determine whether Orai1 affects the SK3–CaM binding site, we investigated SK3–CaM FRET in the absence compared to the presence of Orai1. Remarkably, the presence of Orai1 led to a significant reduction in FRET of CFP-SK3 with both YFP- CaM_WT_ as well as YFP-CaM mutants (CaM_MUT_, CaM_1,2_) (Figure 3G,H).

In accordance with these findings, the K^+^ currents of the almost inactive SK3 V544W mutant were also partially recovered upon the co-expression of CaM, which were not further enhanced by Orai1. Furthermore, while CaM_MUT_ left the SK3 V544W mutant almost inactive, an additional co-expression of Orai1 also partially restored the activation of this SK3 pore mutant (Appendix A).

In summary, we showed that Ca^2+^-mediated gating of SK3 channels is established via constitutively bound CaM which requires the N-lobe to be intact for Ca^2+^ binding. Abolished SK3 channel activity due to excess of the CaM_MUT_ was restored in the presence of overexpressed Orai1, suggesting that Orai1 is able to compensate for disabled CaM function. The stimulatory action of Orai1 requires an intact CaM binding site within SK3. Restoration of SK3 currents in the presence of Orai1 in conjunction with reduced interaction of CaM_WT_, as well as CaM mutants with SK3 in the presence of Orai1, suggests that Orai1 competes, either directly or allosterically, with CaM for the SK3 CaM binding site to boost K^+^ currents.

### 3.4. Both Orai1 N- and C-Termini Play a Crucial Role in the Interplay of Orai1 and SK3

In the following, we investigated the molecular determinants within Orai1 required to maintain the interplay with SK3 using physiological SK3 solution conditions with [EGTA]_intra_.

Initially, we investigated two prominent loss-of-function Orai1 mutants, Orai1 K85E [37,72] and Orai1 L174D [73], known to interfere with the formation of a functional Orai1 pore [42], in the presence of the SK3 channel. Co-expression of either mutant with the SK3 channel revealed similar K^+^ currents as seen for cells expressing SK3 alone, but they were significantly lower compared to those for SK3 in the presence of Orai1-WT (Figure 4A–D).

Confocal fluorescence microscopy exhibited comparable co-localization of SK3 with Orai1, Orai1 K85E, and Orai1 L174D (Figure 4E). In control experiments, we used two other Orai1 mutants (Orai1 L81K K85E and Orai1 E173K), that have been recently reported to restore or maintain an opening-permissive Orai1 conformation [42]. As expected Orai1 L81K K85E and Orai1 E173K retained the ability to enhance SK3 K^+^ currents to a similar extent to Orai1-WT (Figure 4A–D). Thus, in accord with the Orai1 E106Q pore mutant, other mutations negatively affecting the Orai1 pore geometry interfere with the interplay with SK3.

Furthermore, we investigated whether Orai1 N- and/or C-terminal truncations affect the interplay with SK3. It is worth noting that the Orai1 N- and C-terminal deletion mutant, Orai1 79-281, cannot be activated via STIM1 upon store-depletion (Appendix A). Co-expression of Orai1 79-281 with SK3 led to an activation of K^+^ currents to levels comparable to that of SK3 expressed alone, but they were significantly reduced compared to those obtained in the presence of Orai1-WT (Figure 4F,G). Deletion of the N-terminus only (Orai1 N_1-78_), which abolishes [37] STIM1 mediated Orai1 activation, also impaired the enhancement of SK3 mediated K^+^ currents (Figure 4F,G,I). Co-localization of SK3 and Orai1 79-281 or Orai1 N_1-78_ reached comparable levels for SK3 and Orai1-WT (Figure 4H). Interestingly, shorter N-terminal deletion mutants (Orai1 ∆N_1-38/47/64/71/72/74_) (Figure 4I), that have been reported to retain STIM1-mediated Orai1 activation [37], were also fully or partially defective in boosting SK3 K^+^ currents. Only Orai1 N_1-26_ was still able to increase SK3 K^+^ currents (Figure 4I). In line with the effects of the N-terminal truncation mutant, we discovered that point mutants within the extended transmembrane Orai1 N-terminal (ETON) region (aa 72-90), known to abolish STIM1-mediated activation, also interfere with the enhancement of SK3 K^+^ currents (Figure 4J). Additionally, the C-terminus is required for an interplay with SK3, as the C-terminal deletion mutant Orai1 1-281, which shows significantly reduced STIM1-mediated Orai1 activation (Appendix A), exhibits a reduced SK3–Orai1 interplay. Interestingly, SK3–Orai1 1-281 co-localization remained unaffected (Figure 4H).

In conclusion, we discovered that Orai1 mediated SK3 K^+^ current enhancements require an intact Orai1 pore conformation in line with our findings with the Orai1 E106Q pore mutants. This also supports our previous findings that the SK3–Orai1 interplay is likely driven by Ca^2+^ entry via Orai1. Moreover, almost the entire Orai1 N-terminus, except for the first 26 N-terminal residues, and the rear part of the C-terminus, are indispensable for the co-regulation with SK3.

### 3.5. STIM1 Is Not Required for, but Can Influence the SK3–Orai1 Interplay Either Inhibitory or Stimulatory

It is well known that the direct binding of STIM1 to Orai1 induces its activation [45,46]. In the following, we investigated whether STIM1 can influence the SK3–Orai1 interplay using physiological SK3 solution conditions with [EGTA]_intra_, unless otherwise stated. Expression of STIM1 when co-expressed with Orai1 and SK3 was confirmed with Western blot (Appendix A).

Initially, we employed an Orai1 C-terminal point mutant, Orai1 L273D, disabled in coupling to STIM1 [35,74], co-expressed it with SK3 and STIM1 and investigated SK3 mediated K^+^ currents. Both in the absence and presence of STIM1, the co-expression of SK3 with Orai1 L273D resulted in comparable K^+^ current activation as with Orai1-WT (Figure 5A–D). This suggests that STIM1 is not required to enhance SK3 currents by Orai1, similar as reported for breast cancer cells [16].

Remarkably, co-expression of SK3, Orai1-WT, and STIM1 revealed significantly reduced current activation compared to SK3–Orai1 alone (Figure 1K, Figure 5C,D and Appendix A). Notably, inhibition via the SK channel blocker NS8593 left an inward-rectifying current resembling the CRAC channel current (Figure 1K and Appendix A). Indeed, the latter could be blocked by La^3+^ (Appendix A), suggesting that it arises from STIM1/Orai1 activity (Figure 5F). Moreover, we observed almost no co-localization between SK3 and STIM1 (Appendix A). Interestingly, upon store-depletion via thapsigargin, co-localization of SK3 and Orai1 in the presence of STIM1 was significantly reduced by approximately 50% (Figure 5E,F). In the absence of STIM1, SK3–Orai1 co-localization remained unaltered upon store-depletion (Figure 5F). Vice versa, in the presence of SK3, STIM1–Orai1 co-localization upon store-depletion was significantly reduced by only ~17% (Figure 5G). This suggests that, upon store-depletion, Orai1 favors to move into clusters with STIM1 and away from SK3. Due to the use of physiological solution conditions in our electrophysiological experiments presented in Figure 5A–D, we expect only local Ca^2+^ entry in STIM1/Orai1 clusters (Figure 1F,G). This, together with the fluorescence microscopy studies, might explain the reduced K^+^ currents for Orai1/SK3 containing cells in the presence compared to the absence of STIM1.

Nevertheless, the question arose whether global Ca^2+^ entry due to strong STIM1/Orai1 activation is also able to boost SK3 K^+^ current activation. To investigate this aspect, we enhanced the EGTA concentration in the pipette solution (physiological SK3 solution conditions with [EGTA]_intra_); 200 µM EGTA in the pipette solution resulted in further reduced SK3 K^+^ currents. Higher EGTA concentrations amounting to 300 µM, 500 µM, and 1 mM gradually increased SK3 K^+^ currents (Figure 5H). Remarkably, the enhanced store-depletion potentiates STIM1/Orai1 activation and increases SK3 K^+^ currents. Moreover, while keeping 100 µM EGTA in the pipette solution, but using a higher concentration of 10 mM Ca^2+^ instead of 2 mM Ca^2+^ in the extracellular solution, allowed SK3 K^+^ current activation in STIM1/Orai1/SK3 co-expressing cells to similar levels as observed in cells expressing SK3 and Orai1 (Figure 5I,J), hence abolishing the inhibitory effect of STIM1. Overall, these findings substantiate that, under physiological conditions, STIM1 moves Orai1 away from SK3, and local enhancements in Ca^2+^ concentrations are not sufficient to boost SK3 channel activity. Enforcement of STIM1/Orai1 activation likely allows not only local, but also global Ca^2+^ elevations in the cell, enabling a to boost SK3 K^+^ currents.

To further strengthen this concept, we employed, using 100 µM EGTA in the pipette solution (physiological SK3 solution conditions with [EGTA]_intra_), several well-studied STIM1 mutants that are either constitutively opened (STIM1 L251S [34] and STIM1 1-474 [75]) or locked in the closed conformation (STIM1 Y316S [76,77]; STIM1 L373S [78]; STIM1 L373S, A376S [78]; STIM1 A376K [34]; and STIM1 R426L [34]). We expected that constitutively active STIM1 mutants result in stronger inhibition, while the STIM1 inactive mutants show less or no STIM1-mediated inhibition of SK3 K^+^ mediated currents in the presence of Orai1. Indeed, the constitutively open mutants, STIM1 L251S and STIM1 1-474 [67], showed significantly reduced SK3 K^+^ current activation in the presence of Orai1 compared to those of SK3–Orai1 expressing cells. Co-expression of SK3 and Orai1 with each STIM1 mutant that is locked in the closed state revealed comparable current densities as detected for SK3–Orai1 in the absence of STIM1 (Figure 5K). In agreement with this, we already found reduced co-localization of SK3 and Orai1 in the presence of constitutively active STIM1 mutants before store-depletion, as shown, for example, for STIM1-L251S. Co-localization of SK3 and Orai1 remained unaltered for STIM1 mutants locked in the closed state, as exemplified by STIM1 L373S and STIM1 L373S A376S (Figure 5L,M). Additionally, we tested STIM1 mutants, that couple to Orai1, but impair Orai1 gating (STIM1 400-403, STIM1 L402D, and STIM1_400_AADA_403_) [43]. Indeed, coupling of these STIM1 mutants was sufficient to significantly reduce Orai1-mediated SK3 K^+^ currents (Figure 5K and Appendix A). In accordance with the impairment of Orai1 gating of these STIM1 mutants, we showed that using 1 mM EGTA in the pipette solution instead of 100 µM led to almost completely abolished SK3 K^+^ currents in the presence of Orai1 and STIM1 _400_AADA_403_, contrary to Orai1 and wild-type STIM1 (Appendix A).

In summary, we show that Orai1-mediated SK3 current enhancements can occur independently of STIM1. Nevertheless, under physiological buffer conditions, STIM1 possesses an inhibitory role in SK3–Orai1 current activation, as it moves Orai1 away from SK3. Conversely, reinforcing STIM1–Orai1 activation and subsequent Ca^2+^ entry boosts SK3 K^+^ currents.

### 3.6. SK3 Channels Are Endogenously Expressed in LNCaP Cells

Especially in breast, but also in colon cancer cells, the significance of the SK3–Orai1 interplay for proliferation and migration has already been extensively reported [16,18,19,20]. In prostate cancer, less is known about the potential relevance of the Orai1 and K^+^ channel interplay. Here, we used the LNCaP (Lymph Node Carcinoma of the Prostate) cell line to examine whether SK3 plays a role in the proliferation of prostate cancer cells.

Initially, we investigated the K^+^ current development in LNCaP cells using both, symmetrical and physiological SK3 solution conditions. Interestingly, we recorded I/V relationships typical of SK3 channel currents in particular in cells of passages (P) 3–5, but not earlier or later ones (Appendix A). It is worth noting that K^+^ currents in P3-5 were significantly enhanced by overexpression of Orai1, as shown exemplarily for P4 (Appendix A). To analyze whether these currents are specific for endogenously expressed SK3, we applied diverse SK-sensitive agonists and antagonists. Initially, to confirm the SK3-specific action of diverse reported agonists and antagonists, we applied them to SK3 overexpressing HEK 293 cells. In line with Grunnet et al. [63], 1-EBIO (50 µM), a general activator of SK channels, enhanced SK3 K^+^ currents ~ five-fold (Appendix A). Furthermore, NS309, selective for SK1, SK3, and SK4, also drastically enhanced SK3 K^+^ currents already at a Ca^2+^ concentration of 0.5 µM (Appendix A) in accordance with published studies [79]. Additionally, Cyppa, another selective activator of SK2 and SK3 [62], showed significant K^+^ current enhancements for HEK 293 cells heterologously expressing SK3 (Appendix A). To specifically inhibit SK3 mediated K^+^ currents, we applied NS8593, selectively interfering with SK1, SK2, and SK3 [65]. Indeed, we monitored significant inhibition of SK3 K^+^ currents upon application of 10 µM NS8593 (Appendix A). 4-aminopyridine (4-AP), which has been reported to inhibit SK3, but SK2 channels [63,80], exhibited significant inhibition of SK3 mediated K^+^ currents at a concentration of 10 mM 4-AP (Appendix A).

Using these agonists and antagonists in electrophysiological studies, we then investigated SK3 K^+^ current activation in LNCaP cells under both symmetrical and physiological solution conditions. Indeed, the application of both agonists, NS309 (1 μM) and Cyppa (5 μM) led to strongly pronounced K^+^ currents, particularly in LNCaP cells of passages 3–5. Obtained currents could be reversibly inhibited by the antagonist, NS8593 (10 μM) (Appendix A). Applying the principle of exclusion, one can find that these compounds likely activate and inhibit the activity of SK3 in LNCaP cells (Figure 6 inset). In support, weak expression of SK3 in LNCaP cells, in particular in passage 4, was detected with Western Blot (Appendix A). The band, in the range of 80–90 kDa, matches with the size of SK3 monomers of 81.4 kDa and is in line with the detection of SK3 monomers in cells co-expressing SK3, Orai1, and STIM1 in HEK 293 cells. In agreement with our findings, the recent study by Bery et al. [81] discovered that SK3 is weakly expressed in LNCaP cells, which could be strongly upregulated by Enzalutamide treatment.

Next, we investigated whether there is a potential role of SK3 in proliferation. To determine the effect of the SK3 channel on LNCaP cell proliferation, we performed an MTS assay which allows us to monitor the behavior of cell growth in dependence of above-described agonists (NS309 and Cyppa) and antagonists (NS8593 and 4-AP). The growth of the cells under those conditions was compared to the growth of untreated control cells after 24, 48, 72, and 96 h. The data revealed that 1 μM NS309 led to a significantly enhanced cell proliferation compared to control cells, 96 h after the treatment (Appendix A). Similarly, cells treated with 10 μM Cyppa revealed significantly increased cell viability. The SK3 channel antagonist NS8593 (30 μM) showed significantly reduced cell growth, compared to control cells. Among the antagonists, 4-AP is one of the most selective blockers of SK3 channels. While 3 mM 4-AP already resulted in significant inhibition, 10 mM almost completely inhibited the cell growth (Figure 6A,B and Appendix A).

Additionally, STIM1 and Orai1 have been found to be expressed in LNCaP cells and to facilitate store-operated Ca^2+^ entry [82]. Investigation of the behavior of cell growth depending on Orai1 channel blockers revealed that only 30 µM La^3+^, but not 10 µM La^3+^, reduced LNCaP cell proliferation. Notably, the use of 10 µM La^3+^ together with 30 μM NS8593 further reduced the proliferation compared to the use of only the SK3 blocker NS8593. Intriguingly, among prominent CRAC channel blockers, only BTP2 (10 µM) [83,84] significantly reduced proliferation, while GSK-7975A (100 µM) [85,86] did not alter and Synta66 (100 µM) [87] even significantly enhanced the proliferation of LNCaP cells (Figure 6C,D) as recently also reported for glioblastoma cells by Waldherr et al. [88] and metastatic renal cellular carcinoma by Dragoni et al. [89].

Altogether, these results reveal the presence of the SK3 channel in LNCaP cells, which plays a clear role in cell proliferation. Moreover, also Ca^2+^ entry into LNCaP cells is crucial for proliferation. However, whether this is specific to Orai1 remains unclear. To examine whether the SK3–Orai1 interplay persists also in LNCaP cells, we continued with electrophysiological experiments as described below.

### 3.7. SK3 Mediated Currents Are Promoted by Orai1 in LNCaP Cells

To determine whether Orai1 also increases the K^+^ currents in LNCaP cells, we overexpressed Orai1 or the Orai1 pore mutant E106Q and compared the obtained K^+^ currents with those of non-transfected control cells. Orai1 overexpression revealed typical SK3 mediated K^+^ currents using symmetrical as well as physiological solution conditions, with reversal potentials of ~0 mV and −70 mV, respectively. Orai1 E106Q overexpression led to significantly reduced K^+^ currents when compared to Orai1-WT expressing cells, similar to those of non-transfected control cells (Figure 6E–G and Appendix A). The application of 10 μM NS8593 significantly blocked all recorded K^+^ currents (Figure 6E and Appendix A). As a follow-up, we determined the SK3-mediated K^+^ currents upon overexpression of CaM, CaM_MUT_, or CaM_1,2_ in LNCaP cells, and the results were compared to those obtained in cells overexpressing Orai1-WT. Overexpression of CaM, with or without Orai1-WT, led to a comparable degree of K^+^ current activation as with the overexpression of Orai1 alone (Figure 6H,I), in line with our observations in STIM1/Orai1-DKO HEK 293 cells (Figure 3). Both CaM mutants led to significantly reduced or almost completely abolished currents (Figure 6H–K). In line with our observations in STIM1/Orai1-DKO HEK 293 cells, those abolished currents could be rescued upon Orai1 co-expression. Our results indicate that LNCaP cells endogenously express SK3 channels and that Orai1 similarly promotes SK3–mediated K^+^ currents as in HEK 293 cells. The SK3 K^+^ currents already amplified with Orai1 are not further amplified by the coexpression of CaM, which suggests that the effects of CaM and Orai1 are not additive. Indeed, CaM triggers SK3 channel activation, while CaM_MUT_, in which Ca^2+^ binding is disabled, negatively affects the function of the channel. Interestingly, also in LNCaP cells, Orai1 compensates for inhibitory effects of the CaM_MUT_ and enhances SK3 channel activation.

## 4. Discussion

Previous reports on breast and colon cancer cells [16,18,19,20,90,91,92] reported extensively that their proliferation is driven by a co-regulation of co-localized Orai1 and SK3 channels, while the molecular determinants for this remained unknown. In search of the latter, we discovered in this study that constitutive Ca^2+^ entry via Orai1 boosts SK3 mediated K^+^ currents, which is accompanied by their close colocalization. A crucial site in SK3 for the interplay with Orai1 represents the SK3–CaM binding domain. Within Orai1, an intact pore geometry, as well as intact N- and C-termini, are required to maintain the SK3–Orai1 co-regulation. This SK3–Orai1 interplay was not further promoted by STIM1. Instead, STIM1 can interfere with the co-localization of SK3–Orai1, moving, upon SOCE activation, Orai1 and SK3 apart from each other. In addition to the reported role of the SK3–Orai1 co-regulation in breast [17] and colon [19] cancer cells, we demonstrate using LNCaP cells that it also plays a role in prostate cancer cells.

In this study, we provide evidence that Orai1 and SK3 co-expressing cells exhibit constitutive currents upon low, but not high Ca^2+^ buffering via EGTA. Application of an SK3 channel inhibitor reduced those currents, while a small inward-rectifying current remained, which could be inhibited by La^3+^. This suggests that SK3–Orai1 expressing cells generate small, constitutive Ca^2+^ entry, in line with the previously reported enhanced Ca^2+^ levels in breast cancer cells compared to healthy breast cells, likely due to a unique interplay of SK3 and Orai1 in cancer cells [16].

Generally, SK3 K^+^ currents develop with enhancing cytosolic Ca^2+^ concentrations. We demonstrated that SK3–Orai1 co-expressing cells show increased K^+^ current activation compared to cells expressing SK3 only. Enhancements in SK3 K^+^ currents in the presence of Orai1 likely arise from Ca^2+^ entry across Orai1. Indeed, using nominally Ca^2+^ -free or a divalent cation-free, Na^+^ containing solution at the extracellular side abolished Orai1 mediated SK3 K^+^ current enhancements. SK3 K^+^ currents were gradually enhanced with increasing Ca^2+^ concentrations of the extracellular solution in the presence of Orai1. Moreover, different EGTA concentrations in the pipette solution always led to higher K^+^ current activation in cells containing SK3 and Orai1, compared to cells expressing SK3 only. Enhanced Ca^2+^ buffering via increasing EGTA concentrations from 200 to 500 μM enabled activation of SK3 K^+^ currents only in the presence of Orai1. Furthermore, 100 µM BAPTA instead of 100 µM EGTA in the pipette solution did not allow significant enhancements of SK3 K^+^ currents in the presence compared to the absence of Orai1. Consistently, not only the application of La^3+^ blocks enhancement of Orai1 mediated SK3 K^+^ currents, but also the application of the CRAC channel blocker GSK-7975A, as well as the co-expression of the prominent pore mutant Orai1 E106Q. Analogously, loss-of-function mutants, known to interfere with proper pore hydration and maintenance of an intact pore geometry (e.g., Orai1 K85E and Orai1 L174D) [37,42,93], impair Orai1 induced SK3 K^+^ current enhancement. Therefore, the presence of Orai1 is a prerequisite for the pronounced K^+^ current activation via SK3. High buffering of cytosolic Ca^2+^ or impaired Orai1 activity interferes with the positive feedback mechanism of the SK3–Orai1 interplay. Collectively, these results clearly indicate that SK3 channel activation is likely triggered via Ca^2+^ influx through Orai1 channels.

Within SK3, its CaM binding site emerged as crucial for the interplay with Orai1. Generally, SK channel currents enhance with increasing intracellular Ca^2+^ concentrations in a CaM dependent manner, as previously demonstrated [50,51,53,54,64,70]. Here, we further show that the Ca^2+^ dependent activation of SK3 is driven by constitutively bound CaM, requiring two functional Ca^2+^ binding sites at the first and second EF hand. CaM mutants deficient in Ca^2+^ binding (CaM_1,2_ or CaM_MUT_) affect SK3 channel activity in an inhibitory manner. Orai1 did not further enhance SK3 K^+^ currents driven by overexpressed CaM, demonstrating that the effects of Orai1 and CaM on SK3 are less likely additive. Remarkably, our data yield that the inhibitory effect of one of these CaM mutants can be partially overruled by Orai1. Interestingly, Orai1 reduces the interaction of both, CaM_WT_ and CaM mutants with SK3, suggesting that Orai1 boosts SK3 channel activity, via either direct or indirect competition with the CaM binding site within the SK3 channel. In line with this, SK3 mutants which are defective in CaM binding cannot be activated by Orai1. The activation of an SK3 pore mutant could be partially restored by Orai1 or CaM, likely due to its still-intact CaM binding site. Overall, these results suggest that Orai1 enhances SK3 K^+^ currents via an interplay with the SK3–CaM binding site, either directly or allosterically. Interestingly, diverse agonists, such as 1-EBIO [94,95,96], NS309 [79] and Cyppa [62], have also been reported to bind to the interaction interface of the CaM N-lobe and the SK channel [97,98]. In addition, PIP_2_ has been shown to modulate SK2 channel activity via binding to the interface of CaM and the SK C-terminus [99].

Important regions within Orai1 that maintain the co-regulation with SK3 but are dispensable for their co-localization represent almost the entire N- and the C-termini, in addition to an intact pore region. Additionally, for the BKCa channel, it was recently shown that both cytosolic termini of Orai1 are important for their interplay [11]. In regard of the SK3–Orai1 interplay, only the first 26 amino acids in the Orai1 N-terminus are not required for the Orai1-mediated SK3 K^+^ current enhancements. It is worth noting that the first part of the Orai1 N-terminus (aa 1–63) has been reported to include sites involved in cAMP-mediated phosphorylation and caveolin binding [100,101,102,103,104,105], and thus it might be involved in stabilizing the SK3–Orai1 interplay. Future investigations are required to determine the involvement of this region in the SK3–Orai1 interplay.

Furthermore, the enhancements of SK3 K^+^ currents in the presence of Orai1 can occur independently of STIM1 in line with previous reports [16,20,52,57]. Accordingly, Orai1 boosts SK3 K^+^ currents also in STIM1/Orai1-DKO HEK 293 cells. In addition, disturbed STIM1–Orai1 coupling sites in either STIM1 (e.g., STIM1 L373S) or Orai1 (Orai1 L273D) do not interfere with the enhancement of K^+^ currents.

Nevertheless, STIM1 is able to affect the SK3–Orai1 interplay in an inhibitory manner under physiological ionic conditions. The reason for this is that, in STIM1/Orai1/SK3 co-expressing cells, Orai1 favors interplay with STIM1 upon store-depletion as revealed by patch-clamp and co-localization experiments. In support of this, our pool of critical STIM1 mutants in a quiescent or open conformation maintain or inhibit the SK3–Orai1 co-regulation, respectively. It is possible that the expression level of STIM1, Orai1 and SK3 in different cell types determines whether either STIM1–Orai1 or SK3–Orai1 co-regulation becomes decisive.

Aside from physiological ionic conditions, enhanced EGTA concentrations in the pipette solution or increased Ca^2+^ concentrations outside the cell lead STIM1 to even stimulate K^+^ currents in Orai1 and SK3 overexpressing cells. Herein, robust STIM1/Orai1 Ca^2+^ influx enhances SK3 mediated K^+^ currents likely due to an increase in global Ca^2+^ levels within the cell. This is in line with the recent observation for BKCa K^+^ currents which could be further stimulated by STIM1-mediated Orai1 activation [11]. Under physiological solution conditions, STIM1 mediated Orai1 activation probably only allows local Ca^2+^ elevations. Due to the separation of STIM1-Orai1 complexes from SK3 channels, no Ca^2+^ influx mediated K^+^ current activation can be established.

In addition, our data indicate that the SK3–Orai1 interplay also drives LNCaP cell proliferation. Our pharmacological findings, together with Western blot studies, suggest that SK3 is endogenously expressed in LNCaP cells in line with Bery et al. [81]. Furthermore, by using different SK channel drugs, we were able to narrow down that both SK3 and Orai1 are crucial for LNCaP cell proliferation. In line with our results in the standard overexpression system, we were also able to show in LNCaP cells that Orai1-WT promoted SK3 currents, while Orai1 E106Q failed to increase the current density. Additionally, CaM mutant-induced inhibition of SK3 K^+^ currents was restored by Orai1. This further supports our hypothesis that an interplay of Orai1 and SK3 is crucial for LNCaP cell function and proliferation.

In future investigations, it yet must be determined whether and how accessory proteins establish the Orai1–SK3 interplay, stabilize the close co-localization of SK3 and Orai1, and/or assist STIM1 to move Orai1 away from SK3. There is considerable evidence that STIM1 and Orai1 activation is modulated via a series of other regulatory proteins [106,107,108]. Some prominent ones represent SigmaR1, SPCA2, and Caveolin [17,56,109]. SPCA2 has been reported to induce the constitutive activity of Orai1 independently of STIM1 in breast cancer cells [110]. In two separate studies, SigmaR1 has been demonstrated to interplay with STIM1 [111] and SK3 [56]. In breast and colon cancer cells, it has been shown that the SK3–Orai1 co-regulation occurs in cholesterol-rich regions [16,19], indicating a potential role of cholesterol or Caveolin in determining the regulation of the SK3–Orai1 interplay [19,112,113,114,115].

## 5. Conclusions

Collectively, we discovered that close co-localization of SK3 and Orai1 enhances K^+^ current activation likely via local enhancements of Ca^2+^ levels across Orai1. Mechanistically, Orai1 competes with CaM, the typical SK channel activator, to increase SK3 activation via the CaM binding site. In addition, the SK3–Orai1 interplay requires both the Orai1-N- and the C-terminus but occurs independently of STIM1. Interestingly, STIM1 can disrupt their co-regulation by removing Orai1 from SK3. We demonstrate that the synergy of Orai1 and SK3 occurs not only in HEK 293 cells, but also exists in prostate cancer cells, where it likely regulates the function and growth.

## Figures and Tables

**Figure 1 cancers-13-06357-f001:**
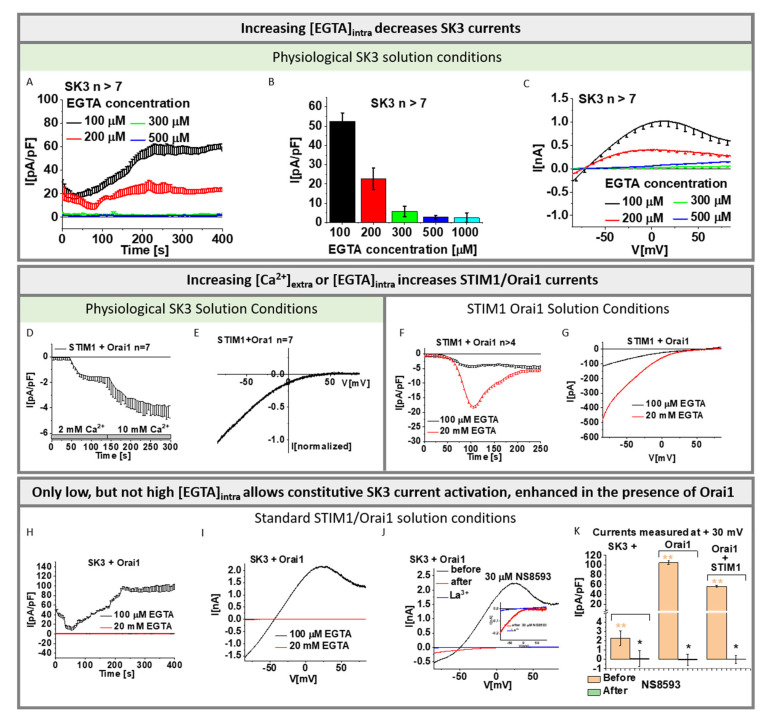
Characterization of SK3 and STIM1/Orai1 channel currents. (**A**) Time course of SK3 mediated whole—cell outward K^+^ currents at +30 mV using physiological solution conditions and recorded in the presence of different EGTA concentration in the pipette solution; (**B**) Block diagram depicts maximum current densities measured in (**A**); (**C**) Current–voltage relationship (I/V) corresponding to (**B**); (**D**) Time course of whole—cell inward currents at −74 mV of Orai1 in co-expression with STIM1 under physiological solution conditions. Inward currents activated upon passive store—depletion via 100 µM EGTA are shown in 2 mM followed by 10 mM extracellular Ca^2+^ solution; (**E**) I/V relationships corresponding to (**D**) in 10 mM extracellular Ca^2+^ solution; (**F**) Time courses of whole—cell inward currents at −74 mV of Orai1 in co-expression with STIM1 under standard STIM1/Orai1 solution conditions. Inward currents activated upon passive store—depletion via 100 μM EGTA compared to 20 mM EGTA are shown; (**G**) Respective I/V traces corresponding to (**F**); (**H**) Time course of SK3 + Orai1 mediated K^+^ currents recorded using standard STIM1/Orai1 solution conditions with either 100 μM or 20 mM EGTA in the pipette solution; (**I**) The I/V relationship corresponding to (**H**); (**J**) I/V relationship of SK3 + Orai1 mediated K^+^ currents recorded using standard STIM1/Orai1 solution conditions and 100 μM EGTA in the pipette solution before (black) and after (red) addition of SK channel blocker NS8593 (30 μM) and subsequent application of La^3+^ (10 µM). (inset) The remaining current upon NS8593 (red) displaying inward rectifying behavior was blocked by La^3+^ (blue). (**K**) The block diagram shows the current densities before and after application of SK channel blocker NS8593 for SK3, SK3 + Orai1, and SK3 + Orai1 + STIM1 *n* > 5 measured at +30 mV. The Mann–Whitney test was used for statistical comparison considering differences statistically significant at *p* < 0.05. The asterisk (*) highlights the statistical significance (*p* < 0.05) before and after application of SK channel blocker NS8593 to currents of SK3, SK3 + Orai1, and SK3 + Orai1 + STIM1 expressing cells. The asterisks (**), as also indicated by corresponding color (light orange), highlight the statistical significance (*p* < 0.05) between the currents recorded upon individual transfections of SK3, SK3 + Orai1, and SK3 + Orai1 + STIM1. All experiments in figure were performed in normal HEK 293 cells.

**Figure 2 cancers-13-06357-f002:**
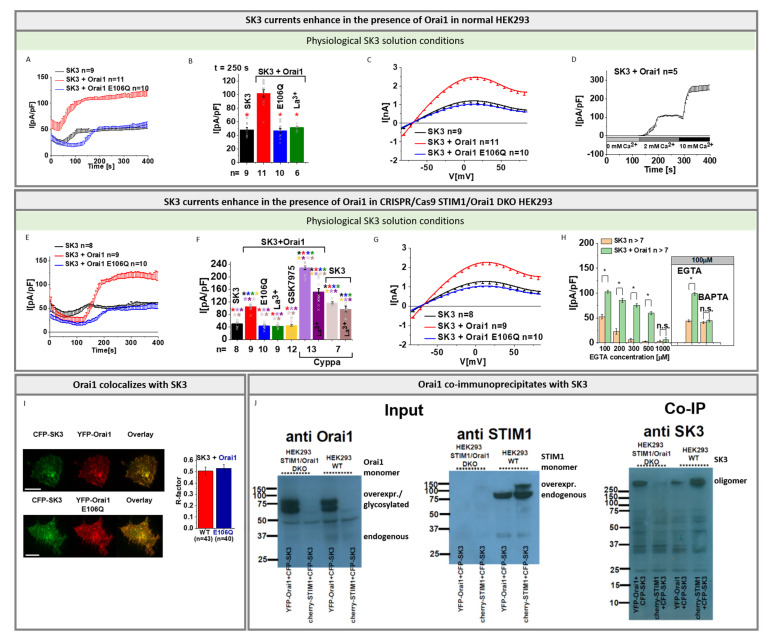
Orai1 enhances SK3 channel K^+^ currents. (**A**) Time course of K^+^ currents of SK3 expressing compared to SK3 and Orai1 or Orai1 E106Q co-expressing HEK 293 cells under physiological solution conditions. Pipette solution contains 100 µM EGTA and 144 mM K^+^ and bath solution contains 2 mM Ca^2+^ and 5 mM K^+^; (**B**) Block diagram with maximum current densities at *t* = 250 s corresponding to (**A**) and Orai1 + SK3 currents upon application of 10 μM La^3+^. The Mann–Whitney test was used for statistical comparison considering differences statistically significant at *p* < 0.05. The asterisk (*), as also indicated by the corresponding color (red), highlights the statistical significance of the currents recorded upon individual transfections of SK3, SK3 + Orai1, SK3 + Orai1 E106Q, SK3 + Orai1 upon application of 10 μM La^3+^. The currents of SK3, SK3 + Orai1 E106Q and SK3 + Orai1 upon application of 10 μM La^3+^ are significantly reduced when compared to SK3 + Orai1; (**C**) The I/V relationship of maximum currents measured in (**A**); (**D**) Time course experiment of HEK 293 cells expressing SK3 + Orai1 upon application of 0 mM Ca^2+^ and subsequent switch to 2 mM Ca^2+^ followed by 10 mM Ca^2+^ solution; (**E**–**G**) Experiments identical to (**A**–**C**) performed in STIM1/Orai1 DKO HEK 293 cells, (**F**) includes in addition SK3 + Orai1currents upon application of 10 µM GSK-7975A, 5 µM Cyppa and 5 µM Cyppa + 10µM La^3+^ and SK3 currents upon application of 5 µM Cyppa and 5 µM Cyppa + 10µM La^3+^. The Mann–Whitney test was used for statistical comparison considering differences statistically significant at *p* < 0.05. The asterisks (*), as also indicated by corresponding colors (black, red, blue, green, yellow, light purple, dark purple, light brown, dark brown), highlight the statistical significance of the currents recorded upon individual transfections of SK3, SK3 + Orai1, SK3 + Orai1 E106Q, SK3 + Orai1 upon application of 10 μM La^3+^, 10 µM GSK-7975A, 5 µM Cyppa and 5 µM Cyppa + 10µM La^3+^ and SK3 currents upon application of 5 µM Cyppa and 5 µM Cyppa + 10µM La^3+^. The asterisk of the particular color above the bar indicates the significance of *p* < 0.05 to the corresponding individual bar of the same color, respectively, which is applicable for each bar; (**H**) Maximum K^+^ currents of SK3 and SK3 + Orai1 expressing HEK 293 STIM1 DKO cells in response to different EGTA concentrations (100, 200, 300, 500, and 1000 µM) or 100 µM BAPTA in the pipette solution under physiological solution conditions. The Mann–Whitney test was used for statistical comparison considering differences statistically significant at *p* < 0.05. The asterisks (*), as also indicated by corresponding color (black), highlights the statistical significance of the currents recorded upon different EGTA or BAPTA concentration upon overexpression of either SK3 or SK3 + Orai1; (**I**) Co-localization studies in STIM1/Orai1 DKO HEK 293 cells performed with a pixel-by-pixel analysis and the corresponding block diagram showing the comparison of YFP-Orai1 with CFP-SK3 and YFP-Orai1 E106Q with CFP-SK3 (scale bar: 10 μm); (**J**) Western blots of wild-type and STIM1/Orai1 DKO HEK 293 cells upon overexpression of either SK3 and Orai1 or SK3 and STIM1 showing either the input of Orai1 detected with an anti-Orai1 antibody, input of STIM1 detected with an anti-STIM1 antibody, or co-immunoprecipitation of Orai1 and SK3 or STIM1 and SK3 detected with an anti-SK3 antibody. While experiments in (**A**–**D**) were performed in normal HEK 293 cells, those in (**E**–**G**) were performed in STIM1/Orai1 DKO HEK 293 cells. Western blots and Co-IP were performed in both normal and STIM1/Orai1 DKO HEK 293 cells as indicated.

**Figure 3 cancers-13-06357-f003:**
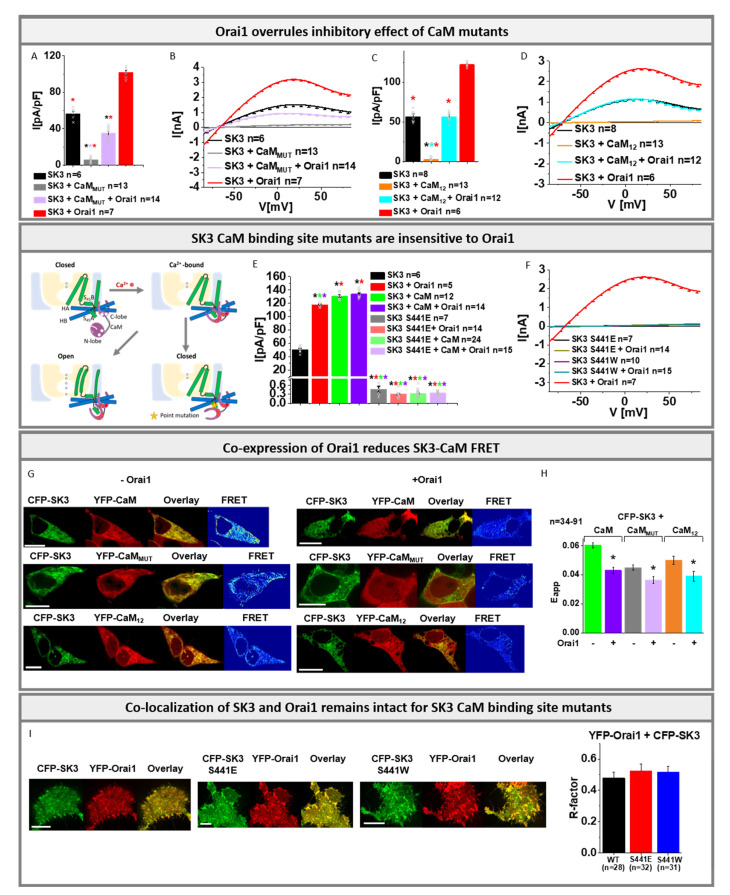
Orai1 overrules inhibitory effect of CaM mutants: (**A**) Block diagram with maximum current densities measured in STIM1/Orai1 DKO HEK 293 cells upon co-expression of CaM_MUT_ with SK3 channel in the absence or presence of Orai1 in comparison to cells containing SK3 or SK3 + Orai1. The Mann–Whitney test was used for statistical comparison considering differences statistically significant at *p* < 0.05. The asterisks (*), as also indicated by corresponding colors (black, red, light purple), highlight the statistical significance of the currents recorded upon individual transfections of SK3, SK3 + CaM_MUT,_ SK3 + CaM_MUT_ + Orai1, SK3 + Orai1. The asterisk of the particular color above the bar indicates the significance of *p* < 0.05 to the corresponding individual bar of the same color, respectively, which is applicable for each bar; (**B**) I/V relationship corresponding to (**A**); (**C**) Block diagram with maximum current densities measured in STIM1/Orai1 DKO HEK 293 cells upon co-expression of CaM_12_ with SK3 channel in the absence or presence of Orai1 in comparison to cells containing SK3 or SK3 + Orai1. The Mann–Whitney test was used for statistical comparison considering differences statistically significant at *p* < 0.05. The asterisks (*), as also indicated by corresponding colors (black, red, cyan), highlight the statistical significance of the currents recorded upon individual transfections of SK3, SK3 + CaM_12,_ SK3 + CaM_12_ + Orai1, SK3 + Orai1. The asterisk of the particular color above the bar indicates the significance of *p* < 0.05 to the corresponding individual bar of the same color, respectively, which is applicable for each bar; (**D**) I/V relationship corresponding to (**C**); Scheme represents the proposed structure of the single subunit of the SK channel with constitutively bound CaM. The subunit consists of 6 TM domains with the pore region located between the fifth and sixth segments. The opening mechanism of the channel is illustrated. Upon SK3 CaM point mutation the channel remains in the closed state; (**E**) Block diagram with maximum current densities measured upon expression of SK3 S441E, SK3 S441E + Orai1, SK3 S441E + CaM, and SK3 S441E + CaM + Orai1 compared to SK3, SK3 + Orai1, SK3 + CaM, and SK3 + CaM + Orai1 in STIM1/Orai1 DKO HEK 293 cells. The Mann–Whitney test was used for statistical comparison considering differences statistically significant at *p* < 0.05 The asterisks (*), as also indicated by corresponding colors (black, red, green, purple), highlight the statistical significance of the currents recorded upon individual transfections of SK3 S441E, SK3 S441E + Orai1, SK3 S441E + CaM, and SK3 S441E + CaM + Orai1 compared to SK3, SK3 + Orai1, SK3 + CaM, and SK3 + CaM + Orai1. The asterisk of the particular color above the bar indicates the significance of *p* < 0.05 to the corresponding individual bar of the same color, respectively, which is applicable for each bar; (**F**) I/V relationship of SK3 S441E, SK3 S441E + Orai1, SK3 S441W, and SK3 S441W + Orai1 compared to SK3 + Orai1; (**G**) Image series depict YFP-CaM with CFP-SK3 in STIM1/Orai1 DKO HEK 293 cells compared to YFP-CaM_MUT_ or YFP-CaM_12_ with CFP-SK3, overlay and pixelwise calculated N_FRET_ index for a representative cell (scale bar: 10 μm) in the absence (left) and presence (right) of Orai1; (**H**) Bar graph diagram depicting FRET of heterologously overexpressed CaM and SK3 in comparison to CaM_MUT_ or CaM_12_ and SK3 in the presence or absence of Orai1. The asterisk highlights the statistical significance (*p* < 0.05). (**I**) Co-localization studies in STIM1/Orai1 DKO HEK 293 cells performed with a pixel-by-pixel analysis and the corresponding block diagram showing the comparison of YFP-Orai1 with CFP-SK3, YFP-Orai1 with CFP-SK3 S441E, and YFP-Orai1 with CFP-SK3 S441W (scale bar: 10 μm). All experiments in figure were performed in STIM1/Orai1 DKO HEK 293 cells.

**Figure 4 cancers-13-06357-f004:**
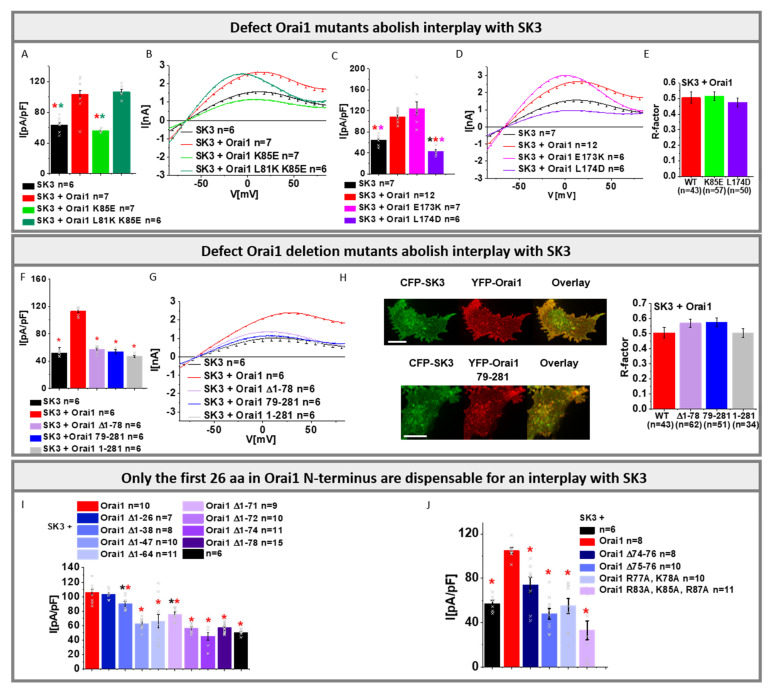
Critical sites within Orai1 that mediate co-regulation with SK3. (**A**) Block diagram with maximum current densities measured in CRISPR/Cas9 STIM1/Orai1 DKO HEK 293 cells upon co-expression of Orai1 K85E or Orai1 L81K K85E with SK3 channel in comparison to cells containing SK3 or SK3 + Orai1. The Mann–Whitney test was used for statistical comparison considering differences statistically significant at *p* < 0.05. The asterisks (*), as also indicated by corresponding colors (red, green), highlight the statistical significance of the currents recorded upon individual transfections of SK3, SK3 + Orai1, SK3 + Orai1 K85E and Orai1 L81K K85E. The asterisk of the particular color above the bar indicates the significance of *p* < 0.05 to the corresponding individual bar of the same color, respectively, which is applicable for each bar; (**B**) I/V relationship corresponding to (**A**); (**C**) Block diagram with maximum current densities measured in STIM1/Orai1 DKO HEK 293 cells upon co-expression of Orai1 E173K or Orai1 L174D with SK3 channel in comparison to cells containing SK3 or SK3 + Orai1. The Mann–Whitney test was used for statistical comparison considering differences statistically significant at *p* < 0.05. The asterisks (*), as also indicated by corresponding colors (black, red, pink), highlight the statistical significance of the currents recorded upon individual transfections of SK3, SK3 + Orai1, SK3 + Orai1 E173K and Orai1 L174D. The asterisk of the particular color above the bar indicates the significance of *p* < 0.05 to the corresponding individual bar of the same color, respectively, which is applicable for each bar; (**D**) The I/V relationship corresponding to (**C**); (**E**) Co-localization block diagram performed in STIM1/Orai1 DKO HEK 293 cells with a pixel-by-pixel analysis showing the comparison of YFP-Orai1 K85E with CFP-SK3, and YFP-Orai1 L174D with CFP-SK3.; (**F**) Block diagram with maximum current densities measured in STIM1/Orai1 DKO HEK 293 cells upon co-expression of Orai1 Δ1-78, Orai1 79-281, or Orai1 1-281 with SK3 channel in comparison to cells containing SK3 or SK3 + Orai1. The Mann–Whitney test was used for statistical comparison considering differences statistically significant at *p* < 0.05. The asterisks (*), as also indicated by corresponding colors (red), highlight the statistical significance of the currents recorded upon individual transfections of SK3, SK3 + Orai1, SK3 + Orai1 Δ1-78, SK3 + Orai1 79-281 and SK3 + Orai1 1-281. The asterisk of the particular color above the bar indicates the significance of *p* < 0.05 to the corresponding individual bar of the same color, respectively, which is applicable for each bar; (**G**) The I/V relationship corresponding to (**F**); (**H**) Co-localization studies performed in STIM1/Orai1 DKO HEK 293 cells with a pixel-by-pixel analysis showing the comparison of YFP-Orai1 79-281 with CFP-SK3 and YFP-Orai1 with CFP-SK3 and the corresponding block diagram showing additionally YFP-Orai1 Δ1-78 with CFP-SK3 (scale bar: 10 μm); (**I**) Block diagram with maximum current densities measured upon co-expression of SK3 with Orai1 Δ1-26/Δ1-38/Δ1-47/Δ1-64/Δ1-71/Δ1-72/Δ1-74/Δ1-78 in comparison to SK3 and SK3 + Orai1. The Mann–Whitney test was used for statistical comparison considering differences statistically significant at *p* < 0.05. The red asterisk represents the significance in relation to SK3 + Orai1, while the black asterisk shows the significance in relation to SK3; (**J**) Block diagram with maximum current densities measured upon co-expression of SK3 with Orai1 Δ74-76, Orai1 Δ75-76, Orai1 R77A K78A, and Orai1 R83A K85A R87A in comparison to SK3 and SK3 + Orai1. The Mann–Whitney test was used for statistical comparison considering differences statistically significant at *p* < 0.05. The red asterisk represents the significance in relation to SK3 + Orai1. All experiments in figure were performed in STIM1/Orai1 DKO HEK 293 cells.

**Figure 5 cancers-13-06357-f005:**
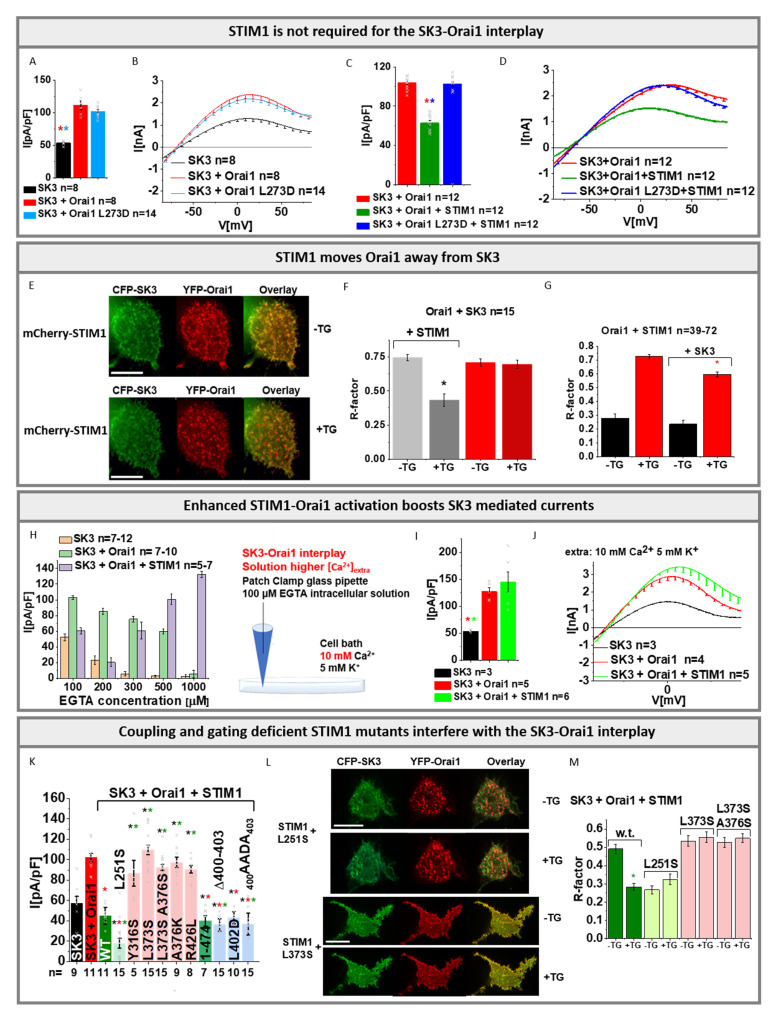
Functional coupling of STIM1 to Orai1 interferes with the SK3–Orai1 interplay. (**A**) Block diagram with maximum current densities measured in STIM1/Orai1 DKO HEK 293 cells upon co-expression of Orai1 L273D with SK3 channel in comparison to cells containing SK3 or SK3 + Orai1. The Mann–Whitney test was used for statistical comparison considering differences statistically significant at *p* < 0.05. The asterisks (*), as also indicated by corresponding colors (red, light blue), highlight the statistical significance of the currents recorded upon individual transfections of SK3, SK3 + Orai1, SK3 + Orai1 L273D. The asterisk of the particular color above the bar indicates the significance of *p* < 0.05 to the corresponding individual bar of the same color, respectively, which is applicable for each bar; (**B**) I/V relationship corresponding to (**A**); (**C**) Block diagram with maximum current densities measured in STIM1/Orai1 DKO HEK 293 cells upon co-expression of Orai1 L273D, SK3, and STIM1 in comparison to cells containing SK3 + Orai1 or SK3 + Orai1 + STIM1. The Mann–Whitney test was used for statistical comparison considering statistically significant differences at *p* < 0.05. The asterisks (*), as also indicated by corresponding colors (red, blue), highlight the statistical significance of the currents recorded upon individual transfections of SK3, SK3 + Orai1 + STIM1, SK3 + Orai1 L273D + STIM1. The asterisk of the particular color above the bar indicates the significance of *p* < 0.05 to the corresponding individual bar of the same color, respectively, which is applicable for each bar; (**D**) I/V relationship corresponding to (**C**); I Co-localization studies with a pixel-by-pixel analysis showing mCherry-STIM1, CFP-SK3, and YFP-Orai1 before and after application of thapsigargin; (**F**) The Pearson correlation coefficient (R-factor) gives a value for co-localizatiI(**E**) before and after thapsigargin treatment. An asterisk (*) indicates a significant difference in co-localization of SK3 and Orai1 before compared to after application of thapsigargin (*t*-test: *p* < 0.05). The control experiment of SK3 + Orai1 in the absence of STIM1 does not reveal a significant difference before compared to after thapsigargin treatment; (**G**) The Pearson correlation coefficient (R-factor) gives a value for co-localization of Orai1 + STIM1 before and after thapsigargin treatment in the absence compared to the presence of SK3 channel. An asterisk (*) indicates a significant difference in co-localization of STIM1 + Orai1 after application of thapsigargin (*t*-test: *p* < 0.05) in the presence of SK3; (**H**) Maximum K^+^ currents of SK3, SK3 + Orai1, and SK3 + Orai1 + STIM1 expressing STIM1/Orai1 DKO HEK 293 cells in response to different EGTA concentrations in the pipette solution (100, 200, 300, 500, and 1000 µM) under physiological solution conditions; (**I**) Block diagram with maximum current densities measured in STIM1/Orai1 DKO HEK 293 cells upon co-expression of SK3 + Orai1 + STIM1 in comparison to cells containing SK3 or SK3 + Orai1 upon application of physiological SK3 solution conditions with high [Ca^2+^]_extra_ of 10 mM (as indicated by the scheme left to the block diagram). The Mann–Whitney test was used for statistical comparison considering differences statistically significant at *p* < 0.05. The asterisks (*), as also indicated by corresponding colors (red, green), highlight the statistical significance of the currents recorded upon individual transfections of SK3, SK3 + Orai1, SK3 + Orai1 + STIM1. The asterisk of the particular color above the bar indicates the significance of *p* < 0.05 to the corresponding individual bar of the same color, respectively, which is applicable for each bar; (**J**) I/V relationship corresponding to (**I**); (**K**) Block diagram with maximum current densities measured in STIM1/Orai1 DKO HEK 293 cells upon co-expression of SK3, Orai1, and STIM1 L251S/Y361S/L373/L373, A376S/A376K/R426L/1-474/Δ400-403/L402D/400AADA403 in comparison to cells containing SK3, SK3 + Orai1, or SK3 + Orai1 + STIM1. The Mann–Whitney test was used for statistical comparison considering differences statistically significant at *p* < 0.05. The red asterisk represents the significance in relation to SK3 + Orai1, while the black asterisk shows the significance in relation to SK3 and the green asterisk indicates the significant difference to SK3 + Orai1 + STIM1; (**L**) Co-localization studies with a pixel-by-pixel analysis showing mCherry-STIM1 L251S or mCherry-STIM1 L373S, CFP-SK3, and YFP-Orai1 before and after application of thapsigargin; (**M**) The Pearson correlation coefficient (R-factor) gives a value for co-localization of (**L**) and additional STIM1 double mutant mCherry-STIM1 L373S co-expressed with CFP-SK3 and YFP-Orai1, all compared to mCherry-STIM1 WT, CFP-SK3, and YFP-Orai1 before and after thapsigargin treatment. An asterisk indicates a significant difference in co-localization of SK3 and Orai1 before compared to after application of thapsigargin (*t*-test: *p* < 0.05). All experiments in figure were performed in STIM1/Orai1 DKO HEK 293 cells.

**Figure 6 cancers-13-06357-f006:**
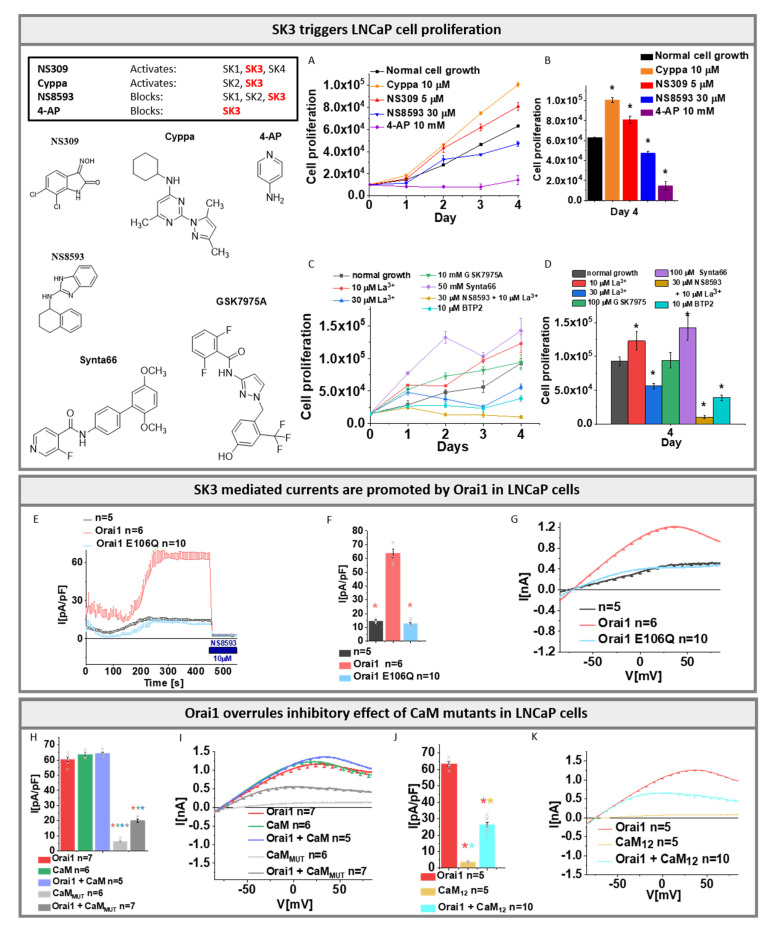
SK3–Orai1 interplay in LNCaP cells: Inset table and chemical structures represent the used agonist and antagonist of SK and Orai1 channels. (**A**) Cell viability of LNCaP cells after 24, 48, 72, and 96 h upon the treatment with SK3 channel agonist 10 μM Cyppa, 5 μM NS309, and antagonist 30 μM NS8593, 10 mM 4-AP detected via MTS assay; (**B**) Block diagram represents the cell proliferation of LNCaP cells after 96 h upon conditions described in (**A**). The Mann–Whitney test was used for statistical comparison considering differences statistically significant at *p* < 0.05. The black asterisk represents the significance in relation to normal cell growth; (**C**) Cell viability of LNCaP cells after 24/48/72 and 96 h upon the treatment with Orai1 channel blocker 10 μM La^3+^, 30 μM La^3+^, 100 μM GSK7975A, 100 μM Synta66, the combination of Orai1 blocker 10 μM La^3+^ with SK3 channel antagonist 30 μM NS8593 and 10 µM BTP2 detected via MTS assay; (**D**) Block diagram represents the cell proliferation of LNCaP cells after 96 h upon conditions described in (**C**). The Mann–Whitney test was used for statistical comparison considering differences statistically significant at *p* < 0.05. The black asterisk represents the significance in relation to normal cell growth; (**E**) Time course of LNCaP currents of endogenously expressed SK3 channel in the absence or presence of Orai1 or Orai1 E106Q. Pipette solution contains 100 µM EGTA and 144 mM K^+^ and bath solution contains 2 mM Ca^2+^ and 5 mM K^+^; (**F**) Block diagram with maximum current densities corresponding to (**E**). The Mann–Whitney test was used for statistical comparison considering differences statistically significant at *p* < 0.05. The light red asterisk indicates the significance to Orai1; (**G**) I/V relationship of maximum currents measured in (**E**); (**H**) Block diagram with maximum current densities measured in LNCaP cells upon co-expression of CaM or CaM_MUT_ with Orai1 in comparison to cells containing Orai1 or CaM_MUT_. The Mann–Whitney test was used for statistical comparison considering differences statistically significant at *p* < 0.05. The asterisks (*), as also indicated by corresponding colors (red, green, purple, grey), highlight the statistical significance of the currents recorded upon individual transfections of Orai1, CaM, Orai1 + CaM, CaM_MUT_ and Orai1 + CaM_MUT_. The asterisk of the particular color above the bar indicates the significance of *p* < 0.05 to the corresponding individual bar of the same color, respectively, which is applicable for each bar; (**I**) The I/V relationship corresponding to (**H**); (**J**) Block diagram with maximum current densities measured in LNCaP cells upon co-expression of CaM_12_ with Orai1 in comparison to cells containing only Orai1 or CaM_12_. The Mann–Whitney test was used for statistical comparison considering differences statistically significant at *p* < 0.05. The asterisks (*), as also indicated by corresponding colors (red, orange, cyan), highlight the statistical significance of the currents recorded upon individual transfections of Orai1, CaM_12_ and Orai1 + CaM_12_. The asterisk of the particular color above the bar indicates the significance of *p* < 0.05 to the corresponding individual bar of the same color, respectively, which is applicable for each bar; (**K**) The I/V relationship corresponding to (**J**). All experiments in figure were performed in LNCaP cells.

## Data Availability

Reported Cancer databases can be found NCI’s Genomic Data Commons (GDC). PanCanAtlas Publications: TCGA PanCancerAtlas; NCI’s Genomic Data Commons: Bethesda, MD, USA. Available online: https://portal.gdc.cancer.gov/ (accessed on 14 December 2020).

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
