# Peer review of "Orai1 Boosts SK3 Channel Activation"

_cancers, 2021, doi:10.3390/cancers13246357_

Round 1
Reviewer 1 Report
In the resubmitted manuscript, the concerns pointed out by the reviewer were almost solved by the addition of the experiments and descriptions. The reviewer has no more concerns.
Reviewer 2 Report
Minor comment
STIM1/Orai1 DKO HEK 293 cells are first mentioned in page 4, but the meaning of DKO is not explained until page 4 (“CRISPR/Cas9 STIM1/Orai1 double knockout (STIM1/Orai1-DKO) HEK 293 cells”). This description needs to be used when the cell line is first mentioned
This manuscript is a resubmission of an earlier submission. The following is a list of the peer review reports and author responses from that submission.
Round 1
Reviewer 1 Report
This is a well written manuscript was a pleasure to read.
Supplementary figure 3G shows the expression of STIM and Orail1 in the wtHEK cells and the double KO cells, however the protein expression levels of SK3, Orai1 and STIM1 in the HEK cells are not shown post-transfection.
The colocalization assessment by immunofluorescence indicates the Orai1 and SK3 co-localise, but it would be good to see this colocalization confirmed using another method such as co-immunoprecipitation.
In the final paragraph of the results the authors state that their results indicated that LNCaP cells have endogenous SK3 channels. It would be good to see a western blot or qPCR showing SK3 expression in the LNCaP cells.
Minor comment
Some of the asterisks to indicate significance are difficult to see e.g. figures 1K, 4I 4J and 5I
Author Response
This is a well written manuscript was a pleasure to read.
We thank the reviewer for the positive evaluation of the manuscript and his/her constructive suggestions.
Supplementary figure 3G shows the expression of STIM and Orail1 in the wtHEK cells and the double KO cells, however the protein expression levels of SK3, Orai1 and STIM1 in the HEK cells are not shown post-transfection.
For our protein overexpression experiments we used fluorescence-labeled proteins – specifically: YFP-Orai1, CFP-SK3 and Cherry STIM1. Thus, protein expression of labeled SK3, Orai1 and STIM1 was verified by confocal fluorescence microscopy (Figure S3J).
Moreover, we verified the expression of overexpressed STIM1, Orai1 and SK3 both in wtHEK and STIM1/Orai1 double KO cells using Western blot. Their expression was detected upon their individual transfection as well as upon their co-transfection (Figure S4 C-D). Interestingly, SK3 is predominantly detectable as a multimer when expressed alone (Figure S4C), while SK3 is predomiantly detectable as monomer upon a triple expression of SK3, Orai1 and STIM1 (Figure S4 D).
The colocalization assessment by immunofluorescence indicates the Orai1 and SK3 co-localise, but it would be good to see this colocalization confirmed using another method such as co-immunoprecipitation.
We agree with the reviewer that co- immunoprecipitation represents a powerful alternative to fluorescence co-localization studies. Thus, we performed co-immunoprecipitation of SK3+Orai1 and included the data in Figure 2 J. In line with our co-localization studies in particular in STIM1/Orai1 DKO cells, Orai1 specifically co-immunoprecipitates with SK3.
In addition, it is worth noting, that observed Orai1-SK3 co-localization is specific to the two channels, as it is reduced in the presence of STIM1 upon TG stimulation (Figure 5 E,F) and also for constitutively active STIM1 mutants (e.g. STIM1 L251S) (Figure 5 K-M), while it does not for inactive mutants (Figure 5 K-M). Moreover, we find a significantly reduced level of co-localization for SK3 and STIM1 compared to SK3 with Orai1 (Figure S8 A).
In the final paragraph of the results the authors state that their results indicated that LNCaP cells have endogenous SK3 channels. It would be good to see a western blot or qPCR showing SK3 expression in the LNCaP cells.
We performed Western blot studies showing weak SK3 expression in the LNCaP cells in particular in passage P4 and included the result in Figure S9 A. The band is the range of 80-90 kD and matches with the size of SK3 monomers of 81,4 kD. Furthermore, these findings are in line with the detection of the SK3 monomer in cells co-expressing SK3, Orai1 and STIM1 (Figure S4D).
Additionally, we screened through recent reports and found literature confirming the presence of SK3 channel in LNCaP cells for example doi: 10.3390/cancers13122947.

Reviewer 2 Report
The manuscript authored by Tiffner et al examines the interplay between the SK3 and Orai1 channels by combining multiple approaches including electrophysiology, mutagenesis and FRET analysis. The data support that Orai1 promotes SK3 K+ currents, via its interplay with the SK3CaM binding site. This interaction occurs independently of STIM1, although STIM1 reduces the extent of SK3-Orai1 co-localization under physiological conditions. Conversely, forced STIM1-Orai1 activity and associated Ca2+ influx promotes SK3 K+ currents.
The article is well-written and provides convincing original data. Overall the study adds to our understanding on SK3 and Orai1 interplay.
The authors are, however, kindly requested to address the following concerns before publication in Cancers:
- Figure 1 : The cells on which the experiments have been performed should be given in the figure legend to help the reader
- Figure 1 : It is mentioned in the legend that the SK3 current was measured at +30mV and the Orai current at -74mV. Why were the measures done at 2 different voltages for SK3 and Orai1 ? Did the authors also measure Orai current at +30 mV ?
- Concerning double KO STIM1/Orai1 experiments performed with HEK cells, Western blot in FigS3G suggests that there is no Orai1 in HEK WT cells. Is it the case, and if so, why was Orai1 knocked-down if it is not expressed ?
- Line 386, it is mentionned that « This suggests that enhancements of SK3 K+ currents in the presence of Orai1 are likely mediated via local Ca2+ entry across Orai1. »
-Did the authors examine local calcium entries through calcium imaging ?
-Have the authors explored the involvement of local calcium entries in the enhancement of SK3 activity ? This may be achieved by analysing the differential effects of EGTA (that should selectively suppress the global Ca2+ signals) and BAPTA (which is expected to suppress both local and global [Ca2+]i elevations) (PMID: 24990931)
- Line 407 it is mentioned that « Remarkably, while SKV544W showed loss of function, Orai1, but not Orai1 E106Q, partially restored its activity ». Could the authors explain how Orai1 can restore activity of SKV544W that is an inactive channel ?
- Concerning the studies with LNCaP cells :
-it is mentioned that LNCaP cell experiments described here, were feasible only within the cell passages 3-5. (Ligne 183). Do the authors have any explanation for this ?
-Figure 6 : the effects of Orai inhibitors on cell proliferation are unclear. Did the authors also test the commonly used BTP2 (also called YM 58483) to inhibit Orai1 mediated effects ?
-Did the authors quantify cell growth in LNCaP cells transfected with dominant negative Orai1 ?
-Concerning the combined effect of NS8593 and La3+, could the authors explain why they chose to combine 2 inhibitors (one for Orai1 and one for SK3) ? Did they also determine if Orai1 inhibition (with La3+ ) blocked SK3 activation (with cyppa for example) ?
Author Response
The manuscript authored by Tiffner et al examines the interplay between the SK3 and Orai1 channels by combining multiple approaches including electrophysiology, mutagenesis and FRET analysis. The data support that Orai1 promotes SK3 K+ currents, via its interplay with the SK3CaM binding site. This interaction occurs independently of STIM1, although STIM1 reduces the extent of SK3-Orai1 co-localization under physiological conditions. Conversely, forced STIM1-Orai1 activity and associated Ca2+ influx promotes SK3 K+ currents.
The article is well-written and provides convincing original data. Overall the study adds to our understanding on SK3 and Orai1 interplay.
We thank the reviewer for the positive evaluation of the manuscript and his/her constructive suggestions.
The authors are, however, kindly requested to address the following concerns before publication in Cancers:
- Figure 1 : The cells on which the experiments have been performed should be given in the figure legend to help the reader
We adapted accordingly. At the end of each figure we stated which cells were used.
- Figure 1 : It is mentioned in the legend that the SK3 current was measured at +30mV and the Orai current at -74mV. Why were the measures done at 2 different voltages for SK3 and Orai1 ? Did the authors also measure Orai current at +30 mV ?
In Figure 1, we show SK3 currents under physiological solution conditions, i.e. at +30 mV, the voltage at which currents are highest. When co-expressed with Orai1, we analysed the currents, both at +30mV (Fig 1K) as well as -74mV (Fig S3E). With this analysis it came clear that when NS8593 was applied to the currents obtained in the presence of Orai1 and SK3, inhibition occurred with only inward currents visible at -74mV, while almost no outward currents were detectable at +30mV. Only subsequent application of La3+ led to full inhibition of the currents also at -74mV, likely due to the inhibition of Orai1 Ca2+ currents. We added a sentence to the methods part that Orai1 currents were analyzed at -74mV (l. 218).
- Concerning double KO STIM1/Orai1 experiments performed with HEK cells, Western blot in FigS3G suggests that there is no Orai1 in HEK WT cells. Is it the case, and if so, why was Orai1 knocked-down if it is not expressed ?
We apologize for the wrong labeling. A slight band for endogenous Orai1 in HEK 293 wt cells is visible around 35 kDa for both, untransfected and pc Orai1 D1-64 transfected HEK 293 cells. This band is not visible in S1/O1 DKO HEK 293 cells. This is now correctly labeled in Figure S4 A.
- Line 386, it is mentionned that « This suggests that enhancements of SK3 K+ currents in the presence of Orai1 are likely mediated via local Ca2+ entry across Orai1. »
-Did the authors examine local calcium entries through calcium imaging ?
We examined for enhanced Ca2+ entry in Orai1 and SK3 co-expressing cells compared to only Orai1 or SK3 expressing cells using Fura-AM in Ca2+ imaging experiments. Unfortunately, we were unable to detect significant enhancements in Ca2+ entry (Figure S3A). Additionally, we investigated for local Ca2+ enhancements using Orai1-GCaMP6 [1] in the presence of SK3 (data not shown), but we were unable to detect enhancements in Ca2+ entry. However, it is probable that the GCaMP6 tethered to Orai1 interferes with proper co-localization with SK3 compared to wild-type Orai1. Overall, with these approaches we assume that we were not able to detect local changes in the Ca2+ concentration.
Interestingly, we observed slightly, although not significantly, enhanced NFAT translocation in Orai1-SK3 co-expressing compared to only Orai1 or SK3 expressing cells (Figure S3 B).
-Have the authors explored the involvement of local calcium entries in the enhancement of SK3 activity ? This may be achieved by analysing the differential effects of EGTA (that should selectively suppress the global Ca2+ signals) and BAPTA (which is expected to suppress both local and global [Ca2+]i elevations) (PMID: 24990931)
As suggested, we compared the effects of 100 µM EGTA and 100 µM BAPTA. We discovered that only in the presence of EGTA, but not BAPTA, SK3 K+ currents are significantly enhanced in the presence compared to the absence of Orai1 (Figure 2H). This supports our hypothesis that in particular local Ca2+ elevations trigger the interplay of Orai1 and SK3.
- Line 407 it is mentioned that « Remarkably, while SKV544W showed loss of function, Orai1, but not Orai1 E106Q, partially restored its activity ». Could the authors explain how Orai1 can restore activity of SKV544W that is an inactive channel ?
We apologize, however, we have to revise our statement in line 478 (originally line 407): Remarkably, while SK3 V544W showed significantly reduced currents compared to SK3 wild-type, Orai1, but not Orai1 E106Q, partially restored its activity. In further investigations, we discovered that not only Orai1, but also CaM is able to partially restore the activation of SK3 V544W (Figure S3 N). A co-expression of Orai1 and CaM led to no additional enhancement in the current, suggesting that their potentiating effect is not additive (Figure S3N). Furthermore, while CaMMUT expression left the SK3 V544W mutant also almost inactive, additional co-expression of Orai1 was able to partially restore the activation of K+ currents (Figure S3 N). These findings are in line with our observations on SK3 wild-type. Mechanistically, we conclude that Orai1 is able to restore the activation of SK3 V544W K+ currents via an interplay with the still intact CaM binding site, in analogy to the mechanism suggested for the co-regulation of SK3 and Orai1.
- Concerning the studies with LNCaP cells :
-it is mentioned that LNCaP cell experiments described here, were feasible only within the cell passages 3-5. (Ligne 183). Do the authors have any explanation for this ?
Interestingly, we discovered typical SK3 like currents which could be activated and inhibited by SK3 channel agonists and antagonists, respectively, only in LNCaP cells of passage 3-5 (P3-P5; Figure 1 -> see attachment). In earlier and later passages, the shape of I/V traces of K+ currents altered and likely resembled that of SK4 channels in line with a recent report by Prevaskaya et al. [2]. Please see examples of the detected currents below.
Figure 1: Current/voltage relationship of whole cell currents in LNCaP cells under symmetrical or physiological solution conditions. Currents are shown for cell of the different passages P2, P4 and P6.
An explanation for this finding might be that SK3 expression is generally weak in LNCaP cells. Indeed, our additionally performed western blot studies showed weak expression of SK3 in cells of P4 (Figure S9A) which is in line with our functional data. The band is the range of 80-90 kDa and matches with the size of SK3 monomers of 81,4 kDa. Furthermore, these findings are in line with the detection of SK3 monomers in cells co-expressing SK3, Orai1 and STIM1 (Figure S4D).
In agreement with our findings, the recent study „Bery et al., Zeb1 and SK3 channel are up-regulated in castration-resistant prostate cancer and promote neuroendocrine differentiation, Cancers, 2021 (doi: 10.3390/cancers13122947)“ discovered that SK3 is weakly expressed in LNCaP cells, which could be strongly upregulated by Enzalutamide treatment.
-Figure 6 : the effects of Orai inhibitors on cell proliferation are unclear. Did the authors also test the commonly used BTP2 (also called YM 58483) to inhibit Orai1 mediated effects ?
Despite Synta66 and GSK-7975A were unable to inhibit LNCaP cell proliferation, the application of BTP2 (10 µM) showed significant inhibition of LNCaP cell proliferation.
It is worth noting, that also other studies [3,4] showed that pharmacological inhibition of Orai1 was unable to block cancer cell (glioblastoma; metastatic renal cellular carcinoma) proliferation. Also in our study, the SK3/Orai1 interplay was not only inhibited by La3+, but to a comparable extent also by GSK-7975A (Figure 2F), despite retained proliferation in the presence of the latter drug. Determining the reason for the maintenance of LNCaP cell proliferation in the presence of CRAC channel inhibitors, although the latter block Orai1-mediated SK3 K+ current, is beyond the scope of this study but will stimulate our future investigations.
-Did the authors quantify cell growth in LNCaP cells transfected with dominant negative Orai1 ?
We tried to investigate whether LNCaP cell proliferation is reduced upon overexpression of the Orai1 E106Q pore mutant. However, the expression level was too low for read-out using the proliferation assay.
-Concerning the combined effect of NS8593 and La3+, could the authors explain why they chose to combine 2 inhibitors (one for Orai1 and one for SK3) ? Did they also determine if Orai1 inhibition (with La3+ ) blocked SK3 activation (with cyppa for example) ?
This combination was chosen because LNCaP cells endogenously express Orai1. We assumed that when the SK3 channel is endogenously expressed, it promotes cell proliferation together with Orai1 even more than the individual channnels. To test this hypothesis we applied not only NS8593 or La3+ separately, but also in combination. Indeed, we discovered that the application of La3+ and NS8593 together led to significantly lower proliferation than in the case of the individual application of the two compounds, which is in support of the SK3/Orai1 interplay.
Moreover we showed that SK3 currents activated by Cyppa were blocked by La3+ only in the presence, but not the absence of Orai1 (Figure 2F).

Reviewer 3 Report
In this manuscript, the experiments were well done. However, the reviewer feels there are no merits for the readers of 'Cancers'. I recommend this manuscript should be submitted to Biophysical Journal and Cell Calcium.
I think the experiments in this manuscript is done using various experimental techniques. So, the reliability of the data is not low. SK3-Orai1 axis is important molecules modulating Ca2+ signaling in some cancer cells. But most work has been done in breast cancer cells, as the authors described in this manuscript. And the most experiments were performed in HEK-293 expression system.In addition, LNCaP cells express KCa1.1 and KCa2.2 but not KCa2.3.
Author Response
In this manuscript, the experiments were well done. However, the reviewer feels there are no merits for the readers of 'Cancers'. I recommend this manuscript should be submitted to Biophysical Journal and Cell Calcium.
We thank the reviewer for the positive evaluation of our experiments.
I think the experiments in this manuscript is done using various experimental techniques. So, the reliability of the data is not low. SK3-Orai1 axis is important molecules modulating Ca2+ signaling in some cancer cells. But most work has been done in breast cancer cells, as the authors described in this manuscript. And the most experiments were performed in HEK-293 expression system.
In addition, LNCaP cells express KCa1.1 and KCa2.2 but not KCa2.3.
We agree with the reviewer that most experiments were performed in HEK293 cells. However, the published studies on breast and colon cancer cells left the molecular determinants controlling the SK3/Orai1 interplay largely unknown. Therefore, we believe that a comprehensive characterization of the key determinants of the SK3/Orai1 interplay in the standard overexpression system HEK293 cells is also of value for cancer cell research, although we are aware that cellular mechanisms might be distinct compared to those in the standard overexpression system. Nevertheless, we were able to show that the molecular mechanisms of SK3/Orai1 co-regulation which we discovered in HEK 293 cells in this study are also valid in LNCaP cells.
Regarding the possible expression of SK3 in LNCaP cells, we would like to point out that we were able to record in particular SK3 channel like K+ currents in cells of the passages P4 and P5, but not in lower or higher passages. Specifically, we were able to enhance K+ currents recorded in P4 and P5 cells by NS309, an activator of SK1, SK3 and SK4, and Cyppa, an activator of SK2 and SK3, and inhibit them by NS8593, an inhibitor of SK1, SK2 and SK3 channels. Thus, we conclude via the principle of exclusion that the main target of these pharmacological compounds is the SK3 channel. Accordingly, the proliferation of LNCaP cells is significantly increased by Cyppa and NS309, while NS8593 induces significant inhibition of LNCaP cells. Moreover, 4-AP, an SK3 but not SK2 blocker, completely inhibited LNCaP cell proliferation. Overall, we conclude that these pharmacological studies indicate that SK3 is expressed in LNCaP cells, as shown for some passages. To identify the reason for the latter will stimulate investigations in our future studies.
To put these pharmacological findings on a more solid basis, we performed further western blot studies showing weak SK3 expression in the LNCaP cells in particular in the passage P4 and included the result in Figure S9 A. The band is in the range of 80-90 kD and matches with the size of SK3 monomers of 81,4 kD. In agreement with our findings, the recent study „Bery et al., Zeb1 and SK3 channel are up-regulated in castration-resistant prostate cancer and promote neuroendocrine differentiation, Cancers, 2021 (doi: 10.3390/cancers13122947)“ discovered that SK3 is weakly expressed in LNCaP cells, which could be strongly upregulated by Enzalutamide treatment.
Furthermore, analogous to our results in HEK 293 cells, we showed that Orai1, but not Orai1 E106Q, is able to boost SK3 channel like K+ currents also in LNCaP cells, which can be specifically blocked by NS8593. Moreover, we discovered in LNCaP cells that the CaMMUT mediated reduction of K+ currents could be restored by Orai1. Thus, we were able to show that the molecular requirements identified in the overexpression system are also valid in the LNCaP cell line, which we believe is also valuable for the journal "Cancers".
